# Life cycle comparison of industrial-scale lithium-ion battery recycling and mining supply chains

Michael L. Machala[1,2,6], Xi Chen [3,4,6,7] ✉, Samantha P. Bunke [3,6], Gregory Forbes[1], Akarys Yegizbay[5], Jacques A. de Chalendar [1], Inês L. Azevedo [1,2], Sally Benson[1,2] & William A. Tarpeh [2,3,7] ✉

Recycling lithium-ion batteries (LIBs) can supplement critical materials and improve the environmental sustainability of LIB supply chains. In this work, environmental impacts (greenhouse gas emissions, water consumption, energy consumption) of industrial-scale production of battery-grade cathode materials from end-of-life LIBs are compared to those of conventional mining supply chains. Converting mixed-stream LIBs into battery-grade materials reduces environmental impacts by at least 58%. Recycling batteries to mixed metal products instead of discrete salts further reduces environmental impacts. Electricity consumption is identified as the principal contributor to all LIB recycling environmental impacts, and different electricity sources can change greenhouse gas emissions up to five times. Supply chain steps that precede refinement (material extraction and transport) contribute marginally to the environmental impacts of circular LIB supply chains (<4%), but are more significant in conventional supply chains (30%). This analysis provides insights for advancing sustainable LIB supply chains, and informs optimization of industrial-scale environmental impacts for emerging battery recycling efforts.

The rise of intermittent renewable energy generation and vehicle electrification has created exponential growth in lithium-ion battery (LIB) production beyond consumer electronics. By 2030, the electric vehicle (EV) sector is projected to dominate LIB growth, accounting for 82% of an estimated 2.4 TWh yr$^{-1}$ of total global LIB production (Supplementary Fig. 1). However, the limited supply of critical materials (e.g., Li, Ni, Co, and Cu[1]) needed for prominent LIB chemistries has exacerbated environmental, economic, national security, and human rights concerns[2,3]. Critical LIB materials are projected to reach major global supply–demand balance deficits before 2030 (Supplementary Fig. 1) without additional investment to improve supply chains. Further, both mining of LIB materials and improper disposal of end-of-life LIBs can damage natural and human

ecosystems, cause occupational hazards during handling, and result in monetary losses[4].

Recycling critical materials in end-of-life LIBs can help alleviate growing environmental concerns and is essential for the long-term sustainability of electrified transportation. While recycled materials may not contribute substantially to global LIB demand for decades, the establishment of domestic circular supply chains is iterative, requiring multiple learning curves as the dominant supply of end-of-life LIB chemistries and form factors evolve and as supply grows. Factors central to the success of recycling include the cost of collecting products, the cost of recycling processes, and the economic value of recovered materials. Considering LIB prices between 2018 and 2021, Li, Ni, and Co comprise the highest embodied economic value (Fig. 1a)[5–12],

[1]Department of Energy Science & Engineering, Stanford University, Stanford, CA, USA. [2]Precourt Institute for Energy, Stanford University, Stanford, CA, USA. [3]Department of Chemical Engineering, Stanford University, Stanford, CA, USA. [4]School of Energy and Environment, City University of Hong Kong, Hong Kong SAR, China. [5]Department of Physics, Kenyon College, Gambier, OH, USA. [6]These authors contributed equally: Michael L. Machala, Xi Chen, Samantha P. Bunke. [7]These authors jointly supervised this work: Xi Chen, William A. Tarpeh. ✉e-mail: xche26@cityu.edu.hk; wtarpeh@stanford.edu

and Al and Cu account for a significant weight percentage of EV battery packs (~25%)[13]. While 99% of lead-acid batteries are recycled in the USA, LIBs exhibit 2–10 times higher economic values but are only recycled 2%–47% globally[14]. The environmental benefits of circularity also strongly motivate LIB recycling given the vast LIB production and emission-intensive mining refinement for key constituent metals. There is a critical need to evaluate the environmental opportunity across several application scales, from numerous small-scale consumer electronic LIBs (e.g., 10–100 Wh) to fewer large-scale transportation and stationary storage LIB packs (e.g., 10–100 kWh)[15]. In addition, the preferred chemistries by automakers have evolved to hedge potential critical mineral shortages and react to market shifts (e.g., increasing emphasis on lithium iron phosphate and sodium-ion batteries), such as the near tripling of lithium carbonate prices in early 2022. Existing LIB variation and supply chain complexity highlight the need for a methodical and comparative life cycle assessment (LCA) between circular (i.e., recycling end-of-life batteries) and conventional supply chains, which is needed for incumbent LIBs today and for prospective recycling strategies with various battery chemistries in the future.

Despite significant progress, current understanding of the environmental impacts of recycling LIBs is still incomplete. The most significant environmental differences between LIB production from circular and conventionally mined cathode materials have not been uniformly attributed to specific supply chain steps which we refer to as extraction, transport, and refinement steps (together referred to as "cradle-to-gate," Fig. 1b). The gate-to-gate refinement processes utilized at established and emerging circular refinement facilities include mechanical separation (Me), pyrometallurgy (Py)[16,17], and

hydrometallurgy (Hy)[18,19]. Specifically, Me physically dismantles LIBs into constituent components, Py leverages elevated temperature to facilitate material transformations, and Hy separates materials in the aqueous phase via leaching, precipitation, and solvent extraction processes. Previous efforts have worked toward calculating environmental impacts (e.g., energy consumption, greenhouse gas emission, and water consumption) of LIB refinement pathways and all cradle-to-gate supply chain steps. However, gate-to-gate analyses of circular refinement processes reported environmental impacts differing by over 30%[18–21] due to inconsistent methodologies. In addition, advancing decision-making capabilities to scale sustainable LIB supply chains requires LCA with more granular data at each step. Incorporating industrial-scale refinement operational data can uniquely inform rational design of refinement technologies. The future development of LIB manufacturing and drivers for a circular battery economy have been projected by academic and industrial researchers[16,22], but industrial-level understanding of the environmental influences of different feedstocks and refinement products is still lacking.

In this study, we quantify the cradle-to-gate environmental impacts of battery-grade cathode material salts manufactured in conventional and circular supply chains across three major steps: material extraction, transport, and refinement (Fig. 1b), focusing on the refinement step. First, we quantify the refinement of mined concentrate from natural deposits into battery-grade materials in conventional supply chains and compare with production of these materials by Redwood Materials (a recycling company in Nevada, USA) in 2021. Two LIB feedstocks are explored: non-energized LIB production scrap from manufacturing facilities and energized end-of-life LIBs collected from consumers. This study contributes insights to inform

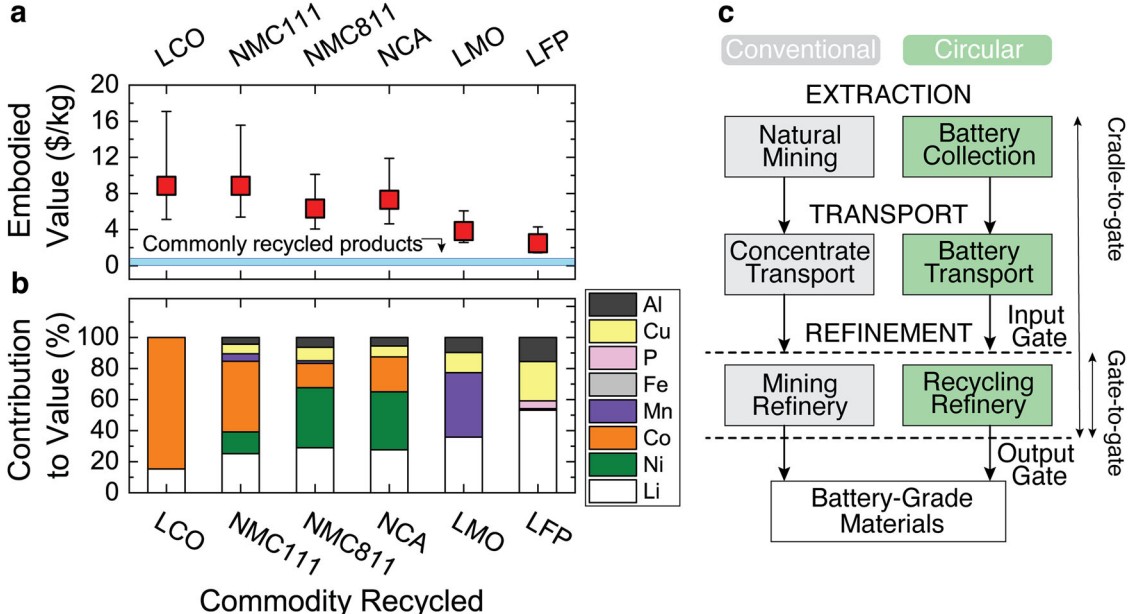

**Fig. 1 | Economic drivers of lithium-ion battery (LIB) recycling and supply chain options for producing battery-grade materials. a** Commodity values of representative LIBs, and **b** relative contributions of embodied metal elements to the LIB values. Representative LIBs are from consumer electronics using lithium cobalt oxide (LCO), and electric vehicle battery packs including lithium nickel manganese cobalt oxide (NMC111 and NMC811), lithium nickel cobalt aluminum oxide (NCA), lithium manganese oxide (LMO), and lithium iron phosphate (LFP). Data are based on market values in 2021 dollars adjusted for inflation between January 2018 and December 2021[5–12], and the uncertainty denotes a 90% confidence interval, which may overlap with the data point in some instances, obscuring their view. The blue shaded area in (**a**) represents the average commodity values of commonly recycled products: glass, paper, plastic, and metal cans (more details are provided in

Supplementary Fig. 1). **c** Cradle-to-gate steps of manufacturing battery-grade LIB materials (i.e., salts) from conventional (gray) and circular (green) supply chains, both of which include three steps: extraction, transport, and refinement. Conventional extraction refers to natural mining, and the circular counterpart is battery collection. Transport in the conventional and circular supply chains move ore concentrate and batteries, respectively. Conventional mining refineries and circular recycling refineries receive ore concentrates and batteries, respectively, and employ different refining technologies. Extraction and transport are considered "upstream steps" relative to gate-to-gate refinement, indicated by the area between "input" and "output" gates. Cradle-to-gate analysis considers the refinement and upstream processes together.

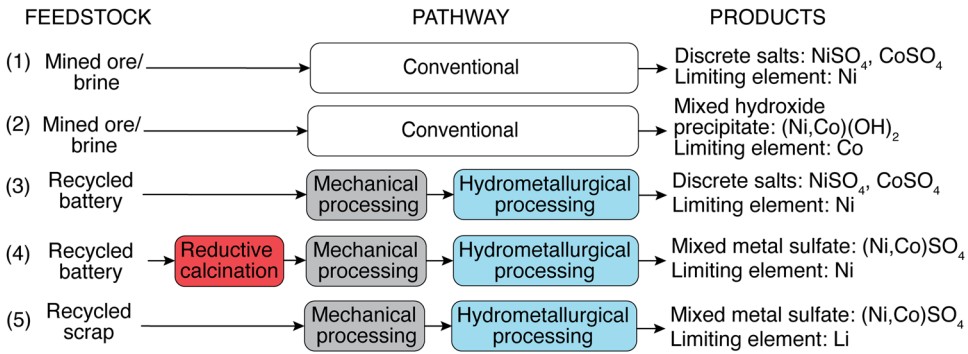

**Fig. 2 | Schematic summarizing feedstocks, pathways, and products in refinement analyses.** Schematic showing the feedstock, pathway, and products as a legend for the refinement methods. Five specific refinement analyses in this study: conventional refining (1 and 2) receives mined ore and brines, and circular refining methods (3–5) recycle from end-of-life batteries or scrap. While all scenarios produce identical $Li_2SO_4$ and $Al_2O_3$, Ni and Co products exist in the form of discrete salts, $NiSO_4$ and $CoSO_4$ (1 and 3), mixed hydroxide $(Ni,Co)(OH)_2$ (2), or mixed metal sulfate $(Ni,Co)SO_4$ (4 and 5). Red, gray, and blue colors denote reductive calcination pyrometallurgical, mechanical, and hydrometallurgical processes.

circular battery manufacturing by addressing three critical gaps in the literature. First, industrial-scale operational data provided by Redwood Materials are analyzed from a more granular level than previous reports and compared to conventional LIB supply chain values based on Argonne National Laboratory's Greenhouse Gases, Regulated Emissions, and Energy use in Technologies (GREET 2021) model[23]. This use of industrial-scale recycling quantitatively identifies the dominant role of input grid electricity in circular refinement at the balancing area level on environmental impact metrics based on industrial-scale LIB recycling data. Second, the influences of the product formats in the circular refinement pathways on environmental impacts are examined by varying the industrial-scale refinement pathways. For both conventional and circular refinement, impacts of producing mixed Ni−Co compounds and discrete salts are analyzed, showing that lower-impact mixed products are worth further investigation. Third, the environmental impacts of upstream processes before gate-to-gate refinement are modeled. The upstream assessment includes the extraction of LIB material from conventional (i.e., mined ore) or circular (i.e., collected batteries) sources and the transport of extracted material to relevant refinement facilities for the production of battery-grade cathode materials as Li, Co, and Ni sulfate or carbonate salts. This upstream modeling shows that refinement is the major contributor to environmental impacts in the circular case, which highlights opportunities to further improve refinement-associated energy and emissions. This study provides LCA insights with primary industrial-scale circular refinement data that includes stepwise, cradle-to-gate comparison of conventional and circular LIB supply chains. With the methodologies and results reported in this study, researchers can prioritize major opportunities to improve process efficiencies, practitioners can benchmark their environmental impacts, and policymakers can incentivize better environmental practices in LIB supply chain management. Granular insights provided by this study based on industrial operation data can also help recyclers optimize the environmental impacts of their refinement processes, and spark more academic-industrial collaborations to further advance the field.

## Results
### Refinement pathways
In this study, analyses of environmental impacts were presented with a focus on the refinement step, followed by analysis of the upstream material extraction and transport steps. In LIB supply chains, the refinement step converts the collected feedstocks into battery-grade salts for further manufacturing (Fig. 2). In both conventional and circular supply chains, the refinement pathways vary significantly depending on multiple factors. Five refinement pathways were compared in this study (Fig. 2). Conventional refinement starts with mined ores/brines (Scenarios (1) and (2) in Fig. 2). Battery scrap generated from manufacturing and assembly is considered a primary recycling source today, and is projected to account for approximately half of the recycling source material in the next decade as battery production outpaces the generation of end-of-life energized batteries[16,22]. Therefore, circular refinement was analyzed starting with either end-of-life batteries ((3) and (4)) or battery scrap (5). Ni and Co in refinement products for subsequent manufacturing can be discrete salts ((1) and (3)) or mixed compounds ((2), (4), and (5)). Target products of the conventional and circular pathways were based on the GREET model and practical recycling operations, respectively. In the following sections, the overall refining environmental impacts were first analyzed, followed by influences of product formats on the refinement step and key contributors to the refining environmental impacts. Lastly, upstream environmental impacts were analyzed and compared to the refinement step. Unless specifically noted, all major analyses were based on 2021 data (data reference years are summarized in Supplementary Table 2).

### Refining lithium-ion batteries lowered environmental impacts
Environmental impacts of refinement pathways in conventional and circular LIB supply chains were analyzed in Fig. 3. The upstream steps of material extraction and transport, which did not have the same granular primary data as refinement, were considered in later sections. Energy consumption, greenhouse gas emissions ($CO_2$-equivalents, $CO_2$-eq; additional criteria air pollutants are detailed in Supplementary Table 3), and water consumption were chosen as key metrics to analyze the environmental impacts of LIB supply chains in this study[19,24,25]. One kg of lithium−nickel−cobalt−aluminum−oxide cathode-equivalent material (NCA-eq) was employed as a functional unit throughout this study for supply chain comparison, accounting for the elemental requirements to produce stoichiometric $LiNi_{0.80}Co_{0.15}Al_{0.05}O_2$. NCA chemistry was selected because it accounted for the second-largest category of EV battery chemistries following NMC batteries in 2021[15,26], and is projected to utilize less Co compared to NMC[14]. Excluding the environmental impacts of material extraction and transport steps, the gate-to-gate production of 1 kg NCA-eq battery-grade material from state-of-the-art conventional mined natural materials consumed 193.9 MJ and 77.3 L $H_2O$ while emitting 14.5 kg $CO_2$-eq (Fig. 3). Refinement of mined material concentrate into battery-grade Ni material dominated NCA environmental impacts, representing >57% of total values. Note that the results were based on GREET 2021 to match the period when circular refinement data were collected at Redwood; more recent environmental impacts of the conventional supply chain

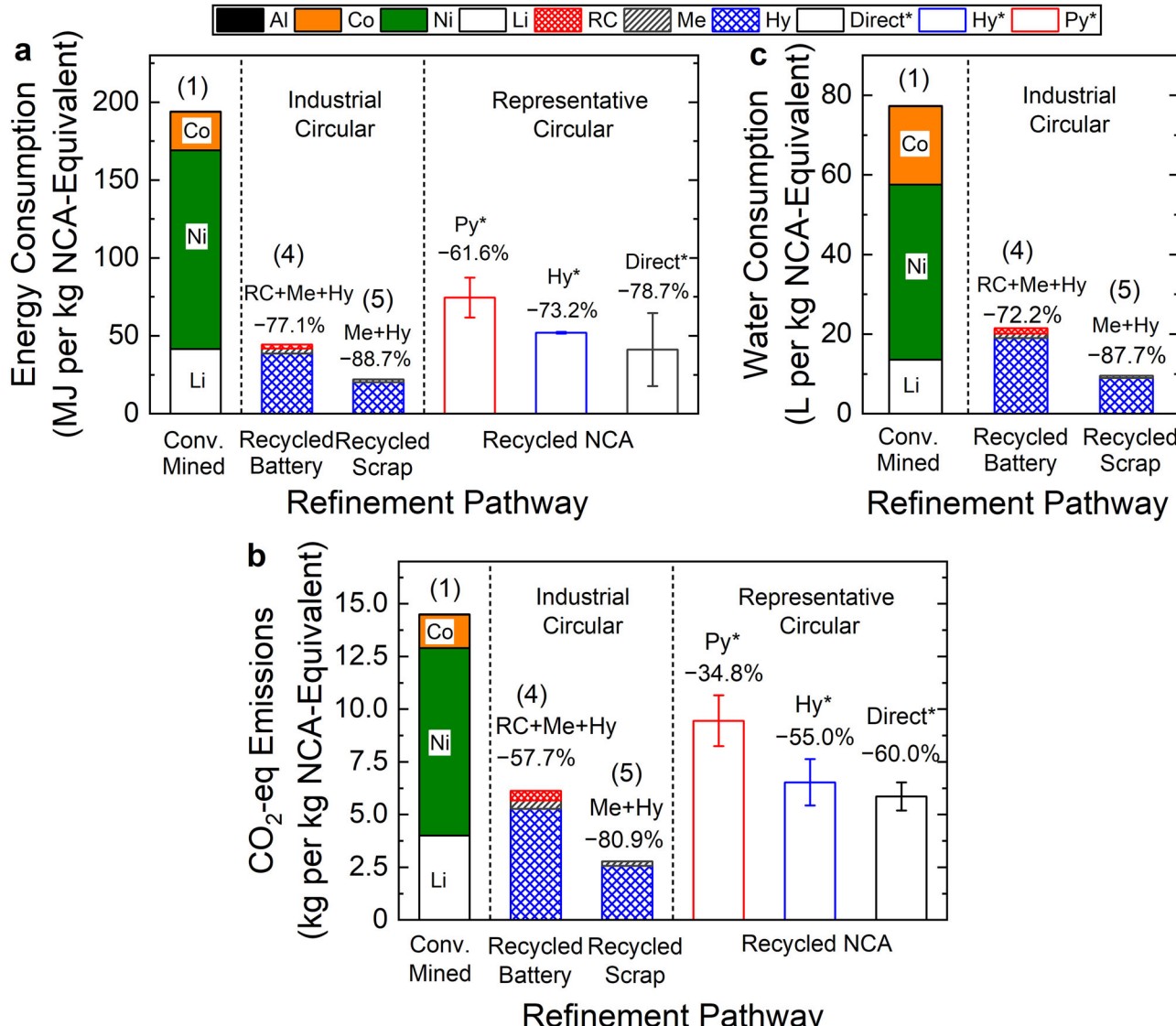

**Fig. 3 | Environmental impacts of conventional and circular refining technologies. a** Energy consumption, **b** $CO_2$-eq emissions, and **c** water consumption of gate-to-gate refinement by different pathways for lithium nickel cobalt aluminum oxide (NCA) battery-grade salts. Numbers in parentheses labeled on the top of stacked bars denote the refinement methods summarized in Fig. 2. The conventional mined pathway (Conv. Mined) refines natural deposits and produces discrete salts (Method (1) in Fig. 2); note that Al is presented on the top of each stacked bar but its contributions are too small to be seen; specific environmental impacts of each element contributor were detailed in Supplementary Table 3. Circular supply chains refine from either mixed energized end-of-life lithium-ion batteries collected from consumers (Recycled Battery, Method (4) in Fig. 2) or non-energized battery scrap from a production facility (Recycled Scrap, Method (5) in Fig. 2), producing mixed metal sulfates. Multi-step circular refinement pathways include mechanical processing (Me, gray), reductive calcination (RC, red), and hydrometallurgy (Hy, blue). RC is an additional processing step for energized batteries and was not used

for non-energized recycled scrap. Open bars in the panels "Representative Circular" denote the environmental impacts of recycling NCA batteries with representative existing pyrometallurgical (Py*), hydrometallurgical (Hy*), and direct recycling (Direct*) methods as comparison, with data obtained from the literature[19]. Literature data were normalized by the same functional unit in this study, and uncertainties were determined by combining two different battery form factors: pouch and cylindrical. The vertical dashed line in each graph demarcates different data types, where the model-based conventional and representative existing pathways were summarized in the left panel, operational data from Redwood Materials were presented in the middle panel ("industrial circular"), and literature data in the right panel ("representative circular"). Note that water consumption has generally not been quantified in previous studies, leading to no literature data panel for (**b**). Environmental impacts of material extraction and transport in the supply chains were not included.

based on the 2023 model were 17.0%–18.3% higher than that of 2021, and are presented in Supplementary Data. The greenhouse gas emissions values were comparable with previous studies based on GREET datasets[23,27] (comparison of environmental impacts with data in literature is detailed in Supplementary Fig. 2d).

The environmental impacts of two circular refinement pathways were presented in each graph in Fig. 3 for mixed-stream LIB feedstocks processed at Redwood Materials: non-energized production scrap from LIB production facilities (recycled scrap) and energized, end-of-

life LIBs collected from consumers (recycled battery). Using a limiting-reagent approach of output products to produce 1 kg NCA-eq material, energy requirements for processing recycled scrap and recycled battery streams were 22.0 MJ and 44.4 MJ per kg NCA-eq materials, significantly lower than conventional refinement by 88.7% and 77.1%, respectively (Fig. 3a). Relatedly, 2.8 and 6.1 kg $CO_2$-eq per kg NCA-eq materials were generated from scrap and battery streams, respectively, a substantial reduction in $CO_2$-eq emissions by 80.9% and 57.7% (Fig. 3b). Water consumption was also lower by 87.7% for scrap and

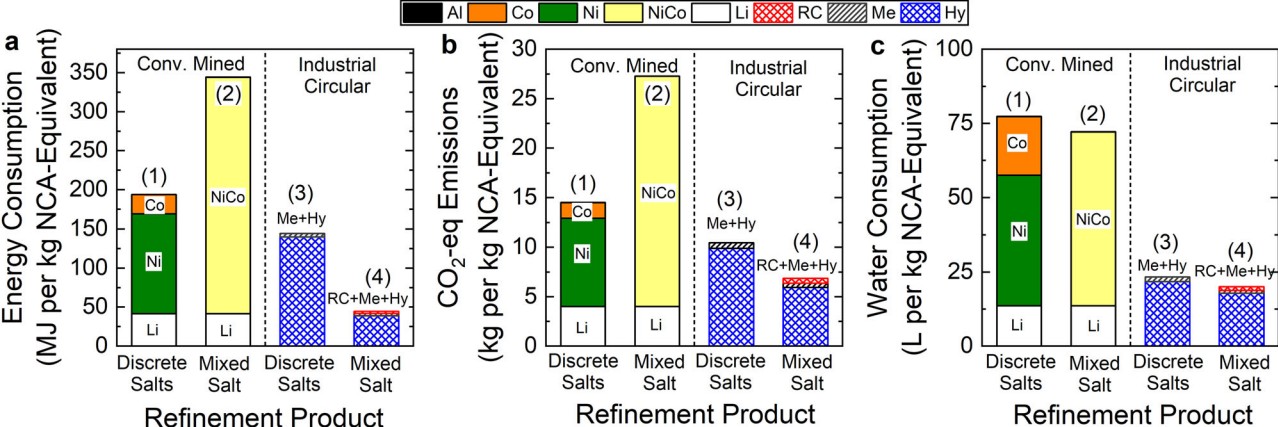

**Fig. 4 | Influences of refining products on environmental impacts in circular refining. a** Energy consumption, **b** $CO_2$-eq emissions, and **c** water consumption. Left and right panels denote conventional (Conv. Mined) and industrial circular pathways that refine end-of-life batteries to discrete Ni and Co salts, or mixed Ni–Co salts. Note that Al is presented on the top of the stacked bars of conventional supply chains but its contributions are too small to be seen (detailed values in Supplementary Table 3). Numbers in parentheses labeled on the top of stacked bars denote the refinement methods summarized in Fig. 2.

72.2% for battery streams relative to the conventional scenario, resulting from the consumption of 9.5 and 21.5 L $H_2O$ per kg NCA-eq materials, respectively (Fig. 3c). Note that while the elemental stoichiometry was identical, the output battery-grade materials varied slightly between conventional ($Li_2CO_3$, $NiSO_4$, $CoSO_4$) and circular ($Li_2SO_4$, $(Ni,Co)SO_4$) refinement in the "Methods" section. Metal sulfates are commonly produced and traded in the battery recycling market[28–31]. Converting the final lithium product to $Li_2CO_3$ did not substantially change the environmental impacts of the circular supply chains (Supplementary Note 3, Supplementary Fig. 2), and impacts of producing discrete or mixed products are examined in the following section.

To produce battery-grade cathode materials, Redwood Materials used a combination of reductive calcination (RC) pyrometallurgical, mechanical (Me), and hydrometallurgical (Hy) LIB refinement processes (pathways detailed in Supplementary Fig. 5). RC is an industrial-scale exothermic pyrometallurgical process that reduces the cathode metal oxide compounds under oxygen-free conditions for subsequent refinement. Unlike dominant industrial pyrometallurgical processes (e.g., direct roasting or smelting) that require high temperature >1400 °C[17,32], the RC process optimizes the working conditions, favoring carbothermal reduction without using graphitic carbon, thus avoiding direct fossil fuel inputs, graphite combustion, and substantial Li loss. Because RC is not required for non-energized LIB production scrap materials, the two feedstock streams (recycled scrap and recycled batteries) were analyzed separately. Energy consumption and $CO_2$-eq emissions of representative existing recycling pathways from the literature, including pyrometallurgy (Py*), hydrometallurgy (Hy*), and direct recycling (Direct*), are also presented in Fig. 3 for comparison. Note that a mechanical processing step is often included in the Hy* refinement. In general, the RC + Me + Hy pathway exhibited comparable energy consumption and $CO_2$-eq emissions with Hy* and Direct* literature values[19], and substantially lower environmental impacts than Py*. Similar to RC, emerging oxygen-free pyrometallurgical processes rely on carbothermic and thermite reduction for recycling cathode metal compounds at moderate temperatures (600–1000 °C)[33–35]. While most carbothermic and thermite reduction processes have been investigated at the lab scale[33–35], our results showed that the RC step accounted for 5.5%–7.5% of the total environmental impacts of the circular refinement step (Fig. 3), demonstrating the environmental feasibility of industrial-scale carbothermal reduction-based pyrometallurgy. Note that RC pyrometallurgy can process energized batteries of varying states of charge, health, and formats with minimal modification, whereas traditional hydrometallurgy often requires discharging energized batteries in a salt bath or removing electrolyte for safe mechanical processing. While this analysis was focused on Redwood Materials refinement pathways, the methodology can be used to evaluate additional refinement pathways (e.g., analysis of a representative hydrometallurgy pathway for energized end-of-life batteries detailed in Supplementary Fig. 2d), or others that use different material feedstocks, refinement processes, and energy supplies.

Among the few studies that directly compare the environmental impacts of circular and conventional NCA refinement using industrial-scale operational data, 35% lower greenhouse gas emissions (Supplementary Fig. 2) were reported for Me + Hy circular refinement compared with the current study[19,27]. However, direct comparison can be inexact due to varying underlying assumptions and data sources. For example, Argonne National Laboratory's GREET and EverBatt models leveraged a combination of technology descriptions from patent applications, literature data on process flow consumptions, industry site visits and surveys, expert advice solicitation, and stated assumptions to form complete pathways. Further, Ciez and Whitacre quantified environmental impacts using output products represented as "metal offsets" for pyrometallurgy or with metals in solution for hydrometallurgy[19] (Supplementary Note 3), rather than cathode salts in this study. In addition, the previous studies included a portion of recycled metal materials in conventional supply chain analysis, whereas this work referenced only mined natural deposits in conventional supply chains to fully deconvolute the environmental impacts[27]. The different conclusions highlight divergent LCA approaches, processing conditions, and the utility of primary industrial data access in addition to modeling processes from literature sources[21].

## Formats of refinement products influenced environmental impacts

Product format is an important factor in understanding and properly comparing LIB refinement pathways (Fig. 2). Ni and Co are key elements for battery manufacturing, and can be traded in the format of mixed metal salts or discrete salt products between battery refiners and battery manufacturers[36,37]. To examine the influences of the refinement product formats, environmental impacts were compared for refinement to mixed salt (e.g., mixed $NiSO_4$ and $CoSO_4$) and refinement to discrete sulfate salts, $NiSO_4$ and $CoSO_4$ (Fig. 4). Both conventional and circular refinement pathways were analyzed.

The GREET model was employed to analyze different conventional mining pathways generating different product formats (detailed in the "Methods" section). In conventional mining, refining to mixed

hydroxide precipitate, $(Ni,Co)(OH)_2$ (Scenario (2) in Fig. 2), increased energy consumption and $CO_2$-eq emissions by 77.% and 89.4%, respectively, over the discrete salts-based pathway (Fig. 4a, b, "Conv. Mined"). The discrete products $NiSO_4$ and $CoSO_4$ are separately produced from Ni-rich and Co-rich ores. In contrast, generating mixed hydroxide salt starts from the Ni laterite ore, which has a low concentration of Co (0.01-0.15%) relative to Ni (0.66-2.4%)[38]. This low Co concentration limits the NCA stoichiometry and increases the total energy cost to generate 1 kg NCA-equivalent materials. On the other hand, water consumption of refining mixed hydroxides was slightly lower (−6.6%) than that of producing discrete salts. In general, the discrete salts-based pathway is favorable for conventional refinement to reduce environmental impacts.

Circular pathways refining batteries to different products were analyzed using the industrial RC + Me + Hy data and the modeling of a representative battery recycling method combining mechanical and hydrometallurgy (Me + Hy, analogous to the Hy* in Fig. 3) refinement (Scenario (3) in Fig. 2). The RC + Me + Hy pathway refines recycled batteries to mixed metal sulfate, $(Ni,Co)SO_4$, whereas the representative Me + Hy produces discrete $NiSO_4$ and $CoSO_4$. Refining into mixed metal sulfate exhibited lower energy consumption (−72.3%), $CO_2$-eq emissions (−41.4%), and water consumption (−8.0%) than the Me + Hy pathway (Fig. 4), because it avoids additional treatment separating $(Ni,Co)SO_4$ to discrete salts. Overall, our results indicated that refining batteries to mixed metal salts instead of discrete salts can substantially save environmental impacts while still satisfying the needs of circular LIB supply chains. Our findings also provide important insights to optimizing plant-scale battery refining operations. In the following sections, mixed salt-based pathways were analyzed for refinement.

### Electricity consumption dominated circular refinement

To further understand the performance limiting factors in the refinement step, the relative environmental impacts of input consumables (e.g., energy, water, commodity chemicals) in the gate-to-gate refinement processes were disaggregated in Fig. 5 (additional criteria air pollutants in Supplementary Tables 10, 11 and Supplementary Figs. 3, 6). Note that the embodied environmental impacts of electricity consumption in Fig. 3 were based on the Nevada Power Company (NEVP) at the Redwood Materials location. Electricity consumption was found to be a principal factor dominating the environmental impacts. For both LIB feedstock pathways (Scenarios (4) and (5) in Fig. 2), electricity accounted for 70.3%–91.0% of the total energy consumption, 70.5%–83.4% of the total $CO_2$-eq emissions, and 56.9%–66.1% of water consumption (Fig. 5a). For both feedstocks, Hy processes comprised the majority of environmental impacts, contributing more than 87.3% to energy consumption, 85.9% to $CO_2$-eq emission, and 88.6% to water consumption. Notably, the additional RC step required for processing energized batteries only marginally contributed to $CO_2$-eq emissions (7.5% of total). Unlike conventional pyrometallurgical processes that require external energy sources[19,32], RC pyrometallurgy is primarily autothermic because it leverages process heat released from exothermic reactions of the LIB materials[39,40]. In addition to electricity consumption, chemical reagents used in circular refinement processes also contributed to embodied environmental impacts. Alkali reagents used to precipitate metals contributed between 7.6% and 19.9% of environmental impacts (largest relative contribution to water consumption). $H_2O_2$ was used to reduce high oxidation state metal compounds for hydrometallurgical leaching of scrap material, and accounted for 10.7%–20.1% of environmental impacts (largest relative contribution to energy consumption).

Because electricity dominated the environmental impacts of LIB recycling processes, we compared several electricity grid balancing areas that emit a range of $CO_2$-eq emissions per MWh (averaged for 2021)[41–43] in Fig. 5b (additional criteria air pollutants detailed in Supplementary Table 12). Substituting NEVP electricity with other

balancing areas including Bonneville Power Administration Transmission (BPAT), California Independent System Operator (CISO), Western Area Power Administration of Colorado-Missouri (WACM), and a 100% Renewable Energy Tariff in Nevada (NV*), yielded a significant reduction in $CO_2$-eq emissions of up to 93.3% (recycled scrap) and 87.4% (recycled battery) relative to conventional refinement (Fig. 5b). Conversely, employing low-carbon electricity grids can increase water consumption compared with NEVP-based operation, following the order of NV* > BPAT > WACM > CISO > NEVP (Fig. 5b). Note that NV*- and BPAT-based circular refinement processes exceeded the water consumption level of conventional refinement due to significant contributions from hydro- and geothermal power. Further investigation into the grid electricity sources of balancing areas revealed a tradeoff between $CO_2$-eq emissions and water consumption based on electricity generation type (Fig. 5c); most electricity sources with relatively low $CO_2$-eq emissions (e.g., those based on bio-, hydro-, or geothermal energy) exhibited high water consumption, and vice versa. This tradeoff also explained the different influences of electricity sources on environmental impacts of the Redwood Materials refinement step and other pathways (detailed in Supplementary Fig. 2d). However, the electricity sources for each balancing area will affect both $CO_2$-eq emissions and water consumption. For example, because NEVP-based electricity includes a relatively large proportion (70%) from $CO_2$-eq emissions-intensive natural gas with low water consumption, a switch to hydro-intensive (73%) BPAT electricity would decrease $CO_2$-eq emissions while increasing water consumption.

### Upstream environmental impacts were lower in circular supply chains

Before the refinement step, LIBs undergo the upstream steps of material extraction and transport to refinement facilities (Fig. 1b). Environmental impacts of these upstream steps were analyzed for two representative LIB chemistries and battery use cases: NCA in EV battery packs, and lithium cobalt oxide ($LiCoO_2$ or LCO) in smartphones. California was chosen to assess circular extraction because it has the largest population and EV market share in the USA[44,45]. Smartphones were considered extracted when collected, aggregated, and transported from all California residents (analyzed per census block) to the nearest existing collection facility (CF)[46]. A shortest-path route for collection was determined by minimizing the distances from block group to CF for the whole state (Fig. 6a; the model is summarized in the "Methods" section and detailed in Supplementary Note 4)[44]. To quantify conventional material extraction environmental impacts from mining, global supply chain data were adapted from GREET[23] (Supplementary Fig. 4, Supplementary Tables 14, 15). Smartphone extraction in the circular supply chain emitted only 0.0186 kg $CO_2$-eq per kg LCO-eq, significantly lower than conventional mining (1.96 kg $CO_2$-eq per kg LCO-eq) by 99.0%. Energy and water consumption were similarly lower in the circular supply chain (Supplementary Table 15).

After extraction, LIB material concentrates transported along domestic and international routes by truck, train rail, and maritime cargo ship to refinery locations (a portion of the network model is presented in Supplementary Fig. 4, and data summarized in Supplementary Tables 4−9, 15, and 16)[23,47–57]. The environmental impacts of transport were quantified by calculating the shortest distance along major transport routes among the participating countries weighted by the relative contributions of the countries to the market for each element (the case of cobalt is presented as an example in Fig. 6b). Details of the modeling method can be found in Supplementary Note 5. Conventional mine-to-refinery environmental impacts were calculated for 1 kg of embodied Li, Ni, Co, and Al metal (Supplementary Table 14). While transport emissions for Li, Ni, and Co ranged from 5.4−6.4 kg $CO_2$-eq per kg embodied metal, Al was three times lower. For the circular case applied to California, smartphones and EV battery packs collected at CFs were transported to a

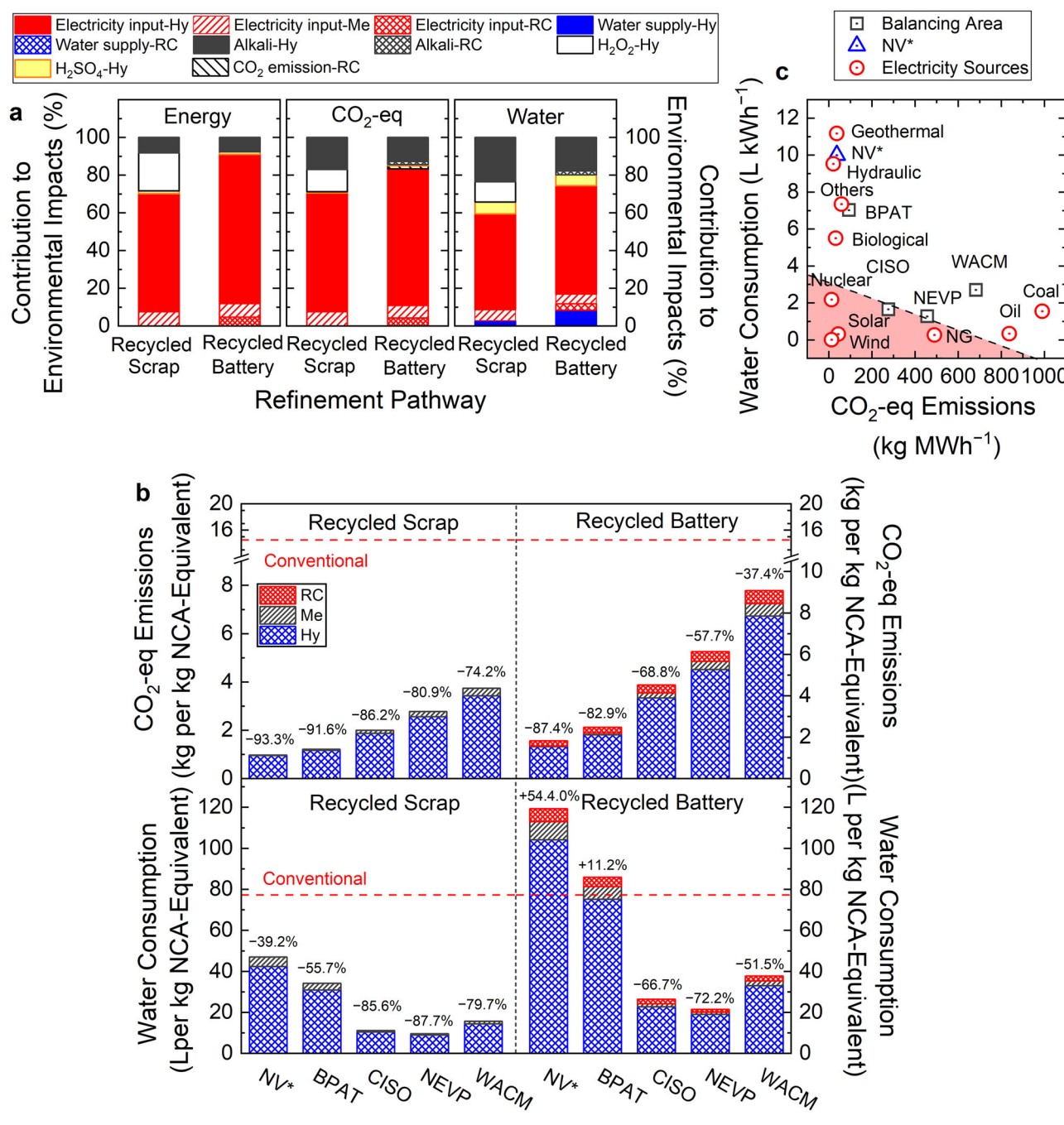

**Fig. 5 | Breakdown of environmental impacts of lithium-ion battery (LIB) recycling using different input electricity sources. a** Contributions to the environmental impacts of recycling processes using electricity from the Nevada Power Company, including energy consumption, $CO_2$-eq emission, and water consumption by different input consumables used in circular processes for LIB feedstocks from production scrap (recycled scrap) and used end-of-life energized batteries (recycled battery) used by Redwood Materials. **b** Environmental impacts of input electricity sources on $CO_2$-eq emissions and water consumption in the LIB recycling operations employed by Redwood Materials methods for production scrap and energized batteries. $CO_2$-eq emissions and water consumption were based on the resources consumed by electricity generated by several electricity sources: Nevada Renewable Energy Tariff (NV*), Bonneville Power Administration

(BPAT), California Independent System Operator (CISO), Nevada Power Company (NEVP), and Western Area Power Administration: Colorado-Missouri (WACM). The red dashed lines denote the environmental impacts of the analogous conventional refining process. Note that influences of energy sources on environmental impacts are only presented for the circular supply chains, but not for conventional supply chains. Specific environmental impacts presented in the figures are detailed in Supplementary Table 12. **c** Tradeoff relationship between embodied water consumption and $CO_2$-eq emission by different power sources, including electricity grids in different locations (circles), purely power sources (squares), and Nevada Renewable Energy Tariff (NV*, triangles). The red dashed line denotes the lower bound of the water-$CO_2$ performance, i.e., the existing electricity grids that have the lowest water consumption and $CO_2$-eq emission simultaneously.

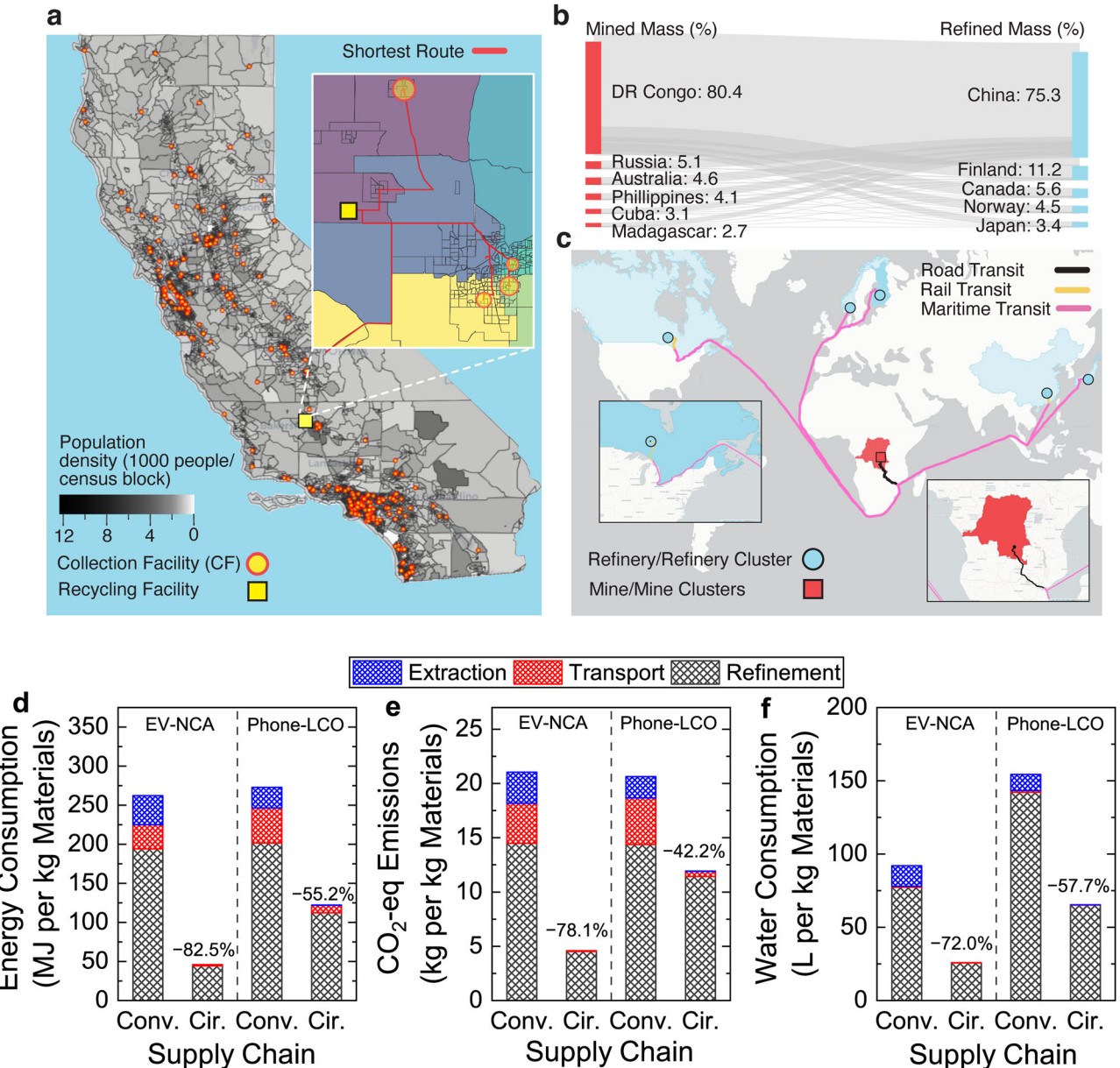

**Fig. 6 | Cradle-to-gate environmental impacts of different supply chains.**
**a** Modeled circular extraction of lithium cobalt oxide (LCO)-based smartphones from every census-block group based on population to the closest existing private or municipal collection facility (CF) using a shortest-route algorithm. Inset details modeled circular transport of smartphones aggregated at CFs and then transported to a central recycling facility at the center (gravity point) of the California population by the shortest route (red lines). Colors of block groups indicate the catchment area of a specific CF, where CF size shows the relative number of smartphones collected in 2021. **b** A weighted distribution estimate of international transport logistics for conventional supply chains between mining and refining countries based on cobalt productivity in the top Sankey diagram. **c** An example of transport logistics for cobalt mined and aggregated in the Democratic Republic of the Congo (DRC) and then shipped via primary road, train rail, and maritime routes

using a shortest distance path to major refinery locations, with insets showing the degree of detail considered. Similar analyses were performed for Li, Ni, Co, and Al. Inserts present more detailed transit routes in DRC and Canada. **d** Energy consumption (left), **e** $CO_2$-eq emissions (middle), and **f** water consumption (right) of conventional (conv.) and circular (cir.) supply chains by supply chain step (material extraction, transport, refinement). NCA-eq cathode used in electric vehicles (EV-NCA, left panels) and LCO-eq cathode material used in smartphones (Phone-LCO, right panels) were provided. Environmental impacts of refinement were analyzed based on electricity generated from balancing grid authority CISO, and upstream supply chain steps (extraction and transport) were based on data from GREET and transport models developed in the preceding section and depicted in (**a**–**c**). Specific environmental impacts of each step were detailed in Supplementary Tables 13–15.

hypothetical central LIB circular refinement facility at the population-weighted center (i.e., gravity point) of California (near Bakersfield)[44]. In conventional supply chains, transporting mined material concentrates accounted for 3.68 kg $CO_2$-eq per kg NCA-eq and 4.32 kg $CO_2$-eq per kg LCO-eq. By comparison, emissions for the transport of aggregated end-of-life NCA EV battery packs (i.e., not disassembled) and LCO smartphone batteries (not separated from

phones) to a circular refinement facility were 0.073 kg $CO_2$-eq per kg NCA-eq and 0.47 kg $CO_2$-eq per kg LCO-eq, 98.2% and 89.1% lower than transport of mined concentrate, respectively. The reduction in $CO_2$-eq emissions was attributed to differences in elemental concentrations of transported materials and aggregate transport distance (e.g., a weighted average of 224 km for circular NCA-eq materials, and 57,600 km for conventional NCA-eq materials).

**Refinement-dominated circular and conventional supply chains**

Combining material extraction, transport, and refinement steps yielded a cradle-to-gate comparison of the differentiated steps of conventional and circular LIB supply chains for producing battery-grade cathode materials (Fig. 6d–f). Here the environmental impacts of the LIB refinement step in California were analyzed for a hypothetical scenario employing the same circular multi-step refinement technologies as the RC + Me + Hy pathway in Nevada, but using California (CISO) electricity to produce battery-grade cathode materials. A circular supply chain in California for NCA EV and LCO smartphone batteries lowered energy and greenhouse gas emissions by at least 47.3% and water consumption by over 42.2%. In the case of recycling NCA EV batteries in California, the entire cradle-to-gate greenhouse gas emissions of the circular supply chain were lower than the transport emissions of mined concentrate in conventional supply chains (Fig. 6d–f and Supplementary Table 15). Circular production of LCO-grade materials led to higher environmental impacts than that of NCA-grade materials based on the mixed-stream feedstock composition analyzed in this study. Note that LCO has relatively lower packing densities of active materials compared with NCA (detailed in Supplementary Table 5, thus elevating the environmental impacts of transporting LCO). Overall, upstream steps (extraction and transport) contributed marginally to the total environmental impacts of both circular supply chains, accounting for ≤4.1% $CO_2$-eq emission, ≤8.2% energy consumption, and ≤0.36% water consumption. Accordingly, the refinement process dominated the environmental impacts of the circular supply chain. In contrast, upstream steps in the conventional supply chain played a larger role (still smaller than refinement) in cradle-to-gate environmental impacts, contributing between 7.8% and 30.4% to the environmental metrics considered (Supplementary Table 15).

## Discussion

This study uses a quantitative cradle-to-gate LCA of disaggregated conventional and circular LIB supply chains and includes primary data from an industrial-scale recycling facility. Various important aspects of the environmental impacts in the refinement step were focused on and analyzed using unit process and operations data from an LIB recycling company, and modeling was employed to examine the environmental impacts of upstream material extraction and transport steps. The analysis revealed that refining end-of-life LIBs into battery-grade cathode materials exhibited lower environmental impacts than conventional refinement of mined materials, mixed salts products were more beneficial for circular refinement, and the source of input electricity is the principal factor governing circular refinement environmental impacts. Upstream circular supply chain steps contribute marginally to overall environmental impacts, and the refinement step comprises the largest source of cradle-to-gate environmental impacts.

Disaggregated analysis of LIB refinement pathways using industrial data provided important insights into the performance and potential of different refinement processes. Pyrometallurgical technologies are advantageous in scalability and operating safety[17,18,58], but are widely considered environmentally intensive due to the high reaction temperature. Oxygen-free carbothermic reduction and thermite reduction have been recently investigated at lab scale for energy-efficient LIB cathode pyrolysis[33–35]. Our analysis showed that industrial-scale carbothermal reduction-based pyrometallurgy (RC) exhibited much lower environmental impacts than prevailing direct roasting and smelting pyrometallurgical and hydrometallurgical pathways (Supplementary Fig. 2). Our findings showed the promise of carbothermal reduction-based pyrometallurgy for pretreatment of end-of-life LIBs for recycling; however, we note that additional work is warranted in controlling process conditions to achieve optimal environmental impacts and product formats without elevating the costs. Existing hydrometallurgy-dominated refinement exhibits advantages over pyrometallurgy in terms of energy efficiency and refinement precision,

but can require costly pretreatment, suffer from limited scalability, and generate secondary waste liquids[59,60]. Our analysis identified chemical consumables such as $H_2O_2$ as important contributors to hydrometallurgy, indicating that environmental impacts of Hy processes can be reduced through more sustainable production methods for chemical inputs (e.g., electrochemical generation of $H_2O_2$)[61]. The alternative direct recycling technology is reported to exhibit comparable environmental impacts to circular refining methods in this study[20], but warrants further assessment after industrial-scale implementation. We note the inconsistencies among existing studies on assessing conventional LIB refinement due to different product compositions, which underscores the need to standardize the functional unit in reporting environmental analyses for future studies in the field. The deviations between our results of circular refinement and model-based literature data highlighted the importance of reconciling models with industrial-scale operating data.

Our findings provide important guidance for material sources and output products in future refinement. Battery scrap is currently the primary recycling source with more gigafactories coming online, but will decrease in the future with quality control advances in manufacturing[11]. While refining end-of-life batteries is more environmentally intensive than refining LIB scrap (Fig. 3), it is critical to improve the technologies for recycling energized batteries when a larger volume of end-of-life batteries becomes available from EVs. Our findings also demonstrated the environmental benefits of mixed metal sulfate refinement products over single salts in the circular supply chain, indicating that the further separations between Ni and Co salts can be avoided.

Electricity greatly influenced environmental impacts in LIB circular refinement, and the variability among grid electricity sources elucidated a tradeoff between $CO_2$-eq emissions and water consumption (Fig. 5). Therefore, considering water consumption and $CO_2$-eq emissions is necessary for selecting recycling facility locations, particularly in water-sensitive or emissions-sensitive scenarios. Further examination suggested that the tradeoff is primarily driven by water-intensive hydroelectric and geothermal electricity in certain locations versus $CO_2$-intensive coal and natural gas in others, implying that increasing the proportion of electricity from nuclear, wind, and solar energy sources simultaneously reduces $CO_2$-eq emissions and water consumption relative to existing balancing areas (Fig. 5).

Analyses of upstream environmental impacts informed more efficient operations for future resource-saving extraction and transport steps. Conventional mining and processing of ore or brine was resource-intensive due to the low natural concentrations of critical materials (0.01%–1%), while critical material concentrations for transport rose to 3%–15% after beneficiation. Further concentrating materials near mine sites or building refineries closer to mine sources can efficiently reduce environmental impacts of the conventional mined materials. In contrast, smartphones contain 5% LCO material by mass, with the batteries themselves at approximately 24% LCO[53]. Circular material extraction via LIB collection decreases environmental impacts by 99% versus conventional. A "shortest-route" approach was used in this study to quantify the environmental impacts of battery extraction and transport supply chain steps. Practical battery collection operations will likely vary based on route selection and preprocessing strategy further influencing environmental impacts[62]. Further investigating the environmental impacts of the disassembly of collected batteries from devices prior to transport can help balance between extraction energy usage and transport emissions (Supplementary Table 5). Trucks are used as the primary vehicle for transport analysis given regulatory concerns that consider LIBs hazardous material in many transportation scenarios[63]. However, alternative transport like railway can further lower environmental impacts by approximately four times versus trucking (Supplementary Table 6), and can be explored for use as aggregation points for long transport combined

with trucking. Upstream process optimization of environmental impacts warrants further investigation, such as the active research area of high-throughput automation of LIB extraction from non-standardized devices and EV battery packs or rapid assessment of LIBs for second life uses. While the current study modeled localized collection and transport inside the state of California, extending the scale to larger regions is critical to further assess the upstream environmental impacts.

As the prevalence of LIBs grows in the mobility sector and beyond, strategic placement of domestic LIB collection, refinement, and manufacturing facilities can further minimize future environmental impacts by considering heterogenous LIB growth by location, collection approach, transportation distance, and electricity source for refinement processes. As LIB production scales, policies informed by consumer surveys, focus groups, pilot testing, and diverse stakeholder engagement will be needed to research and scale battery collection[64]. Business models for collection of all LIB types and sizes will likely vary from manufacturer-led to municipal or private collection programs, and can be influenced by the safety costs while collecting end-of-life batteries as hazardous wastes. In addition to collection costs, the varied scale of collection requires further investigation, particularly for localized environmental impacts. Notably, analogous economic and environmental impacts to local ecosystems of conventional mining are not considered in this analysis, and warrant future studies[65]. Additionally, designing and manufacturing LIBs for recycling in a circular economy can reduce resource usage identified in this study[66]. Employing reusable battery packs can reduce refinement energy and chemical inputs, and designing battery cells and modules favorable for extraction and integration will lower the environmental impacts of the upstream steps. Future efforts should also focus on optimizing refinement processes for subsequent steps of the circular supply chain in LIB manufacturing, product performance, and economic cost.

## Methods

### Goal and scope

The goal of this study was to compare stepwise cradle-to-gate environmental impacts (energy consumption, $CO_2$-eq emission, and water consumption) for two supply chains: a conventional, linear supply chain fed by natural mined material for refinement into battery materials, and a circular supply chain fed by LIBs. Both supply chains produced battery-grade cathode materials. A comprehensive cradle-to-gate analysis of both supply chains considered steps of material extraction, transport, and refinement, and gate-to-gate analysis investigated the refinement step, the focus of this study. A gate-to-gate scope was broadly defined as the boundary surrounding processing facility operations. In this analysis, gate-to-gate refinement only considered direct processing (e.g., alteration, concentration, precipitation) of the feedstock material once it was extracted from its original state and transported to the refinement location (shown in Fig. 1b). For Redwood Materials, this scope included mechanical processing, reductive calcination, and hydrometallurgy (Supplementary Fig. 5). The system boundary did not include other operations outside of the direct refinement processes as discussed in study limitations below.

Two LIB feedstock streams were evaluated: (1) battery production scrap and (2) mixed, spent LIBs from consumers (Supplementary Fig. 5). Upstream of the gate-to-gate supply chain included both material extraction and transport steps. For conventional extraction, GREET was used to quantify the environmental impacts of mining. Transport between the supply chain steps and the circular extraction step was quantified using a logistics transportation model developed in this study, where limitations were summarized below.

### Methodology

An attributional LCA was conducted to quantify and compare conventional and circular LIB supply chains for the production of battery

cathode materials. This analysis complied with the International Organization for Standardization (ISO) 14040 standards but omitted conversion to environmental impact indicators and external review[67]. Data for conventional material extraction (e.g., mining) and refining were adapted from the Argonne National Laboratory's Greenhouse Gases, Regulated Emissions, and Energy Use in Transportation (GREET®) 2021 model. GREET and the ecoinvent 3.3 database[68] were employed for life cycle inventory (LCI) data of chemical consumables for the conventional and circular supply chains.

To assess circular LIB refinement, primary operational data detailing energy, water, on-site emissions, and consumables usage were provided by Redwood Materials and normalized to mass flows of the different elements of interest in input feedstocks and output products. A representative prevailing circular refinement, Method (2) in Fig. 2, was modeled with the software HSC Sim[69], based on the technical procedures available in the literature[58,70–72] and the practical feedstock amount received by Redwood.

Conventional refinement was modeled by aggregating the environmental impacts of the individual refining pathways for each LIB cathode element (Supplementary Table 3), normalizing by the mass of the individual element of interest within the output product (e.g., Li in $Li_2CO_3$) and then normalizing again by the mass of that element in the functional unit for this LCA (defined in the next section). For elements where more than one pathway of production exists in the GREET model (i.e., Ni and Li), the overall environmental impacts were calculated by averaging pathways weighted by their respective share of global production (45% Li production from brine and 55% from ore, and 60% Ni production from mixed hydroxide precipitate and 40% from Class 1 Ni). Both discrete and mixed output products were considered. Discrete salts from conventional refinement were $Li_2CO_3$, $NiSO_4$, $CoSO_4$, and $Al_2O_3$; alternatively, $(Ni,Co)(OH)_2$ was considered as the mixed product. Lithium outputs produced by Redwood Materials were $Li_2SO_4$ (environmental impacts for converting to $Li_2CO_3$ are detailed in Supplementary Note 3), and other outputs existed as mixed metal sulfates of $(Ni,Co)SO_4$ or as $Al_2O_3$ and $Al(OH)_3$. With additional treatment further transforming the mixed metal sulfate into separate Ni and Co compounds, discrete salts as $NiSO_4$ and $CoSO_4$ were analyzed based on modeling of a prevailing Hy + Me refinement pathway. In the cradle-to-gate analysis, material transportation between stages was not included because it was not consistently available in the GREET model. Electricity sources vary between elements, as well as between pathway stages. For example, crude production of $Co(OH)_2$ used a distributed electricity source in the Democratic Republic of the Congo, and the refinement of these materials into $CoSO_4$ and $CoCl_2$ used a distributed electricity source in China. See Supplementary Data for the breakdown of the conventional refining data workflow.

### Defining functional units

Functional units standardize comparisons of the resource consumption and emissions in LCAs. In this study, two functional units were considered to normalize environmental impacts between conventional and circular supply chains: the battery-grade material required to make 1 kg of stoichiometric lithium nickel cobalt aluminum oxide ($LiNi_{0.80}Co_{0.15}Al_{0.05}O_2$, NCA-eq) and lithium cobalt oxide ($LiCoO_2$, LCO-eq) cathode material. Mass was selected as the primary normalizing factor because any energy-based functional unit (e.g., per kWh) could vary based on battery manufacturing and cycling characteristics. NCA chemistry was selected because future cathodes were projected to utilize less Co compared to NMC batteries in EVs[15], and NCA comprised the second-largest category of EV battery chemistries in 2016, following NMC batteries. LCO was a representative chemistry used in handheld rechargeable devices (e.g., cellphones and laptops) which are currently available to recycle in larger quantities than EV LIBs. Environmental impacts of other LIB-relevant materials (Cu and Mn) in conventional supply chains can be found in Supplementary Table 14.

In both conventional and circular supply chains, the extraction, transport, and refinement steps were converted into environmental impacts for the production of battery-grade materials and normalized by NCA and LCO functional units. A limiting-reagent approach was used to quantify the environmental impacts of a functional unit in circular refinement pathways. According to current multi-step pathways using mixed-stream LIB feedstocks (either recycled scrap or recycled battery), the Li output was the limiting element for creating 1 kg of NCA-eq materials from recycled scrap, where other refined elemental products were produced in excess. Relatedly, Ni was the limiting output element from recycled batteries. For multi-step refinement processes, the recovery rate of Ni and Co was 95% and for Li was 92%. Additionally, a sensitivity analysis of environmental impacts from circular refinement was conducted based on facility location in different grid balancing areas and their associated electricity sources.

### Life cycle inventory and assessment

The LCI data for conventional mining pathways were normalized by each critical metal element: Li, Ni, Co, Al, Cu, and Mn (Supplementary Table 14). The LCI data for consumables in the Redwood process were adapted from the GREET 2021 model and ecoinvent 3.3 (Supplementary Table 10)[23,68]. The LCI for the Redwood processes also listed water consumption and criteria emissions for different electricity sources by grid balancing areas in the Western USA (Supplementary Table 12). Three categories of environmental impacts were detailed in this study: energy consumption, air pollutant emissions, and water consumption. Energy consumption included the input electricity for different applications and the energy required to produce required consumables. Criteria air pollutant emissions included the embodied emissions generated by the production of input electricity and the consumed reagents. $CO_2$, $CH_4$, $CO$, $NO_x$, $N_2O$, $SO_x$, $PM_{10}$, and $PM_{2.5}$ were the air pollutants provided in the GREET model and considered here. Greenhouse gas emissions were reported as $CO_2$ equivalents ($CO_2$-eq) summing $CO_2$, $CH_4$, and $N_2O$ weighed by the corresponding 100-year global warming potential (GWP). Water consumption considered withdrawn water not returned to the original source, and both the input city water usage and the embodied water consumption in electricity generation and the manufacturing of consumable materials were included.

### Estimating environmental impacts of material extraction

For conventionally mined ore and brine, energy consumption, $CO_2$-eq emission, and water consumption values were separated for the material extraction processes found in the GREET model. For the circular extraction case, LCO-based smartphones were assumed to be collected and transported to existing private and municipal collection facilities (CFs) from each census-block group in CA, assuming every person owned a cell phone and purchased a new phone every 3 years. A shortest-route method was developed for modeling LCO collection at the closest municipal collection facility. Census blocks centered by their CFs were first determined for the state of California by $k$-means clustering method, and the pathways of transporting LCOs to CFs were identified through minimizing the possible travel distance, realized by Dijkstra's algorithm[73]. Details of the methods are described in Supplementary Note 4.

### Estimating environmental impacts of material transport

In the conventional supply chain, a network model of primary transport routes was established that connects mines to refinery locations for Li, Co, Ni, and Al on a country-level basis (Supplementary Tables 4–9, 15, and 16) because the amount of mined material transported from each mine to each refinery was not known. The distances of the shortest-path routes were calculated between mines and refineries by country, predicated on the closest available modes of transport (including road, rail, and maritime). Details of the modeling method are described in Supplementary Note 5. A major mine cluster or refinery location was selected to represent country-level transport values (Supplementary Tables 7 and 8) based on production volumes, and distances were quantified between international destinations. These distances were used to calculate the energy consumption, $CO_2$-eq emissions, and water consumption associated with the transportation of critical materials as mined concentrate. Mined concentrate referred to ore or brine that was concentrated locally beyond natural concentration values to reduce weight for transport to a refinery. By considering the total elemental mass and elemental weight percentage of the mined concentrate transported along a route (Supplementary Tables 7 and 8), the environmental impacts on a per-element basis were calculated as a global weighted average (Supplementary Table 4) with additional process details in Supplementary Notes 4 and 5.

For the circular case applied to California, end-of-life EV NCA LIBs were aggregated at one CF per county closest to its centroid, where county-level data were the most granular data available. All smartphones were aggregated at their nearest CFs. Aggregated smartphone and EV batteries were assumed transported via truck to a single recycling facility located at the gravity point of California's population based on census block-level data (detailed in Supplementary Note 4). The mass-distances traveled were converted to energy consumption, $CO_2$-eq emission, and water consumption (Supplementary Tables 4 and 15).

### Summary of study limitations

Limitations based on key assumptions of supply chain steps (extraction, transport, refinement) in each supply chain (conventional and circular) are briefly discussed in this section.

Mining data in conventional supply chains in GREET often only refer to one mining country per material, indicating that the global supply chain was difficult to capture. Transport required between mining unit processes (e.g., crushing, flotation, and concentration) prior to refinement was excluded from the current analysis due to limited information in GREET. In collection of end-of-life batteries in smartphones, inefficient transport to a CF (e.g., driving each smartphone individually or taking longer transport routes to a CF) was not considered. In addition, all end-of-life EV battery packs were assumed to be driven to each CF in their original vehicles, which was attributed to the "product use" stage instead of extraction in LCA; therefore, zero $CO_2$-eq emissions were assumed for the extraction step of EV batteries. Batteries may be collected as hazardous wastes, but the potential influences of the safety costs were not considered.

An inter-country LIB material transportation assessment was performed as a weighted distribution between all major mining and refining countries. Results were sensitive to the weight percentage of critical material in transported concentrate presented in Supplementary Tables 7 and 8. Transport between a domestic mine and refinery was not considered, resulting in net zero use of resources in such cases. The resources required to separate an embedded battery from its device prior to a refinement facility were not considered in circular supply chains. Similarly, the effect of transporting only LIBs separated from the devices was not considered. Incorporating the domestic transport and battery separation operations can increase environmental impacts.

Refinement data in conventional supply chains were limited to the country scenarios reported in GREET, and transport between refinement unit processes was not included. Ancillary processes (e.g., transport between unit processes) beyond direct refinement unit processes and embodied resources of the capital equipment used for material refinement were not considered for the circular supply chain. The chemical formats of output products differ between the conventional and circular supply chains, but converting them to the same

products would not substantially change the results due to the similarity between the cathode salts of the two supply chains (Supplementary Note 3). Battery-grade cathode metal sulfates were chosen as the refinement products for the major comparison, differing from the ready-to-use materials for manufacturing, but not substantially influencing the fairness of the comparison.

## Data availability

All data plotted in main and Supplementary Figs. are provided in the Source Data file (.xlsx). Life cycle inventory source data for Figs. 3–5 and all source data for Fig. 6 are provided in the Source Data file, and listed in Supplementary Tables. The mass input and output data of the industrial lithium-ion battery refinement for Figs. 3–5 can be obtained upon request. Data used in building the model for Fig. 6 have been deposited in the OSF database: https://osf.io/zvame/?view_only=8fd188cde196485bbe625b217819242a. Source data are provided with this paper.

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

## Acknowledgements

This work was financially supported in part by StorageX Initiative and Stanford Precourt Institute for Energy.

## Author contributions

M.L.M., X.C., S.P.B., and W.A.T. conceptualized and designed the work. M.L.M., X.C., and S.P.B. contributed equally. X.C., S.P.B., J.A.d.C., and M.L.M. conducted the analysis of the refinement technologies and influences of electricity. M.L.M., G.F., and A.Y. developed the modeling of the upstream steps. X.C., M.L.M., and S.P.B. carried out the interpretation of the data and co-drafted the work. M.L.M., X.C., S.P.B., J.A.d.C., I.L.A., S.B., and W.A.T. discussed the results and commented on the manuscript. X.C. and W.A.T. conducted substantial revisions of the manuscript. W.A.T. supervised the work and secured funding along with S.B. and I.L.A. The submitted and modified version of the manuscript has been approved by M.L.M., X.C., S.P.B., G.F., A.Y., J.A.d.C., I.L.A., S.B., and W.A.T.

## Competing interests

This study was supported by the Stanford StorageX Initiative within the Precourt Institute for Energy. Redwood Materials was consulted for primary industrial data but results were analyzed and interpreted by the listed authors.
