## [Peer Review File · Nature Communications]

REVIEWER COMMENTS

Reviewer #2 (Remarks to the Author):

This study reveals that recycling LIBs into battery-grade materials significantly reduces environmental impacts. The research highlights that electricity consumption is the major factor in the environmental impacts of LIB recycling. It emphasizes the relatively minor contribution of material extraction and transport in circular supply chains, while these steps have a more notable impact in traditional supply chains. The inclusion of primary industrial-scale data from Redwood Materials is an interesting contribution, adding some real-world relevance to the findings. However, some arguments may not be fully supported, and some descriptions of important points are inadequate. The following are the detailed comments and critiques to be addressed:

1. While the study mentions the potential to help recyclers optimize environmental impacts, it would be strengthened by providing more concrete examples or strategies for such optimization. Having Redwood Materials as a case study is interesting, but there are several others using more mature or advanced facilities in US or global markets.
2. The study appears to be well-structured and clear in its objectives and methodologies. However, ensuring that the data sources, assumptions, and limitations are transparently communicated will be crucial for the study's credibility and usefulness. More importantly, the data used in Fig 1 is based on market value from 2018 to 2021, which is quite outdated. In the battery industry, the prices change in weekly or even daily basis, and fluctuation can be large. One typical exam is Li, its prices has changed by about 75% since 2022.
3. The methodology seems robust, utilizing the GREET 2021 model for conventional supply chain analysis and incorporating data from actual industrial operations for the circular supply chain. However, the model is also quite outdated.
4. The results indeed are different from some of the recent findings that transportation can be a significant cost factor, especially collecting spent lithium- ion batteries and transport them as Class 9 hazardous wastes is a huge cost burden. But the authors seem to have not included this factor in discussing recycling as those spent batteries will not automatically concentrate in the recycling facility.
5. The analysis comparing pyrometallurgical and hydrometallurgical processing, and the identification of the environmental benefits of the RC pathway, is valuable. However, the study could benefit from a deeper exploration of the trade-offs between these methods, particularly in terms of scalability and economic viability.
6. The study's focus on upstream impacts, including extraction and transport, is an important aspect that often gets overlooked. The suggestion for further concentrating materials near mine sites or adjusting locations of refineries to reduce environmental impacts is a practical recommendation.

7. The mention of designing and manufacturing LIBs for recyclability is a crucial point, suggesting a shift towards a more sustainable, circular economy. However, this aspect could be explored in more detail, particularly in terms of how current manufacturing practices could be adapted.
8. Give a clearer, more detailed explanation of methodologies, especially where the authors mention modeling and the “shortest route” approach. This provides context and helps readers understand how your conclusions were reached.
9. Where discussing environmental impacts, also consider mentioning the economic aspects more explicitly. Balancing environmental benefits with economic feasibility is crucial for practical applications.
10. The study's broader implications for strategic placement of LIB facilities and policy recommendations are valuable. However, more concrete suggestions or models for policy implementation would enhance the study's practical utility. It would also be beneficial if the study included specific recommendations or identified key areas where policy interventions could be most effective.
11. Expand on the limitations of study and the implications for future research. This not only adds credibility to your work but also guides future studies in this area.

Minor comments:

1. Some phrases might be ambiguous or unclear. For example, line 478: "building reinterests closer to sources" is unclear and might need rephrasing based on the intended meaning.
2. Maintain a consistent verb tense throughout the text. For example, switch from "The analysis reveals" to "The analysis revealed" if the rest of your text is in past tense.
3. Thoroughly proofread the text to catch any typographical errors, misplaced commas, or incorrect word usage.
4. Where you refer to figures (e.g., "Fig. S3", "Fig. 5"), ensure these are easily accessible to the reader and consider including brief descriptions in the text to aid understanding for those who might not have immediate access to the figures.
5. Many references are in random formats.

Reviewer #3 (Remarks to the Author):

This paper presents a comparative life cycle analysis for primary and secondary supply chains for lithium ion batteries (LiBs). The paper is well written, the topic is of interest, the core methodology

is sound, and outcomes relevant for advancing the conversation around designing circular supply chains for LiBs. The major concerns I have about this study have to do with the case for its uniqueness, the significance of some of the key outcomes and the appropriateness of the context for interpreting these results. I think it needs to be updated before considering it for publication. I provide these as comments below for consideration.

Please see attached document for my comments.

REVIEW SUMMARY

This paper presents a comparative life cycle analysis for primary and secondary supply chains for lithium ion batteries (LiBs). The paper is well written, the topic is of interest, the core methodology is sound, and outcomes relevant for advancing the conversation around designing circular supply chains for LiBs. The major concerns I have about this study have to do with the case for its uniqueness, the significance of some of the key outcomes and the appropriateness of the context for interpreting these results. I think it needs to be updated before considering it for publication. I provide these as comments below for consideration.

MAJOR COMMENTS

Lines 128 – 130, 438 – 440: The authors make the case that a distinguishing contribution from this study is the representation of (1) industrial scale circular refinement data, & (2) stepwise cradle-to-gate supply chain comparison, but it is not clear to me what this claim really amounts to.

- For (1), I imagine this has to do with using data from Redwood's process, so what is the value proposition then? Does using data from Redwood's process provide more accurate (or trustworthy) recycling impacts compared to previous studies? Since the authors correctly pointed out that there are significant differences in reported environmental impacts for LiB recycling (Lines 91-94), how does this study address and resolve that problem, and not simply add another data point to literature on LiB recycling impacts?
- For (2), Since several studies have already looked at both primary and secondary battery supply chains and compared the impacts of the various process steps (*even if they did not categorize it as extraction and refining, those steps are implicit in the reference process description*) and across different environmental metrics, I would like more clarity on how and why this work is different, and why that difference is important.

Lines 153 – 154, 286-287, ...: Some of the major study findings, such as that "*recycling exhibit lower environmental impacts than primary production*" or that "*electricity consumption dominates recycling environmental impacts*" have already been established in previous studies. I think the authors need to make a clearer case of what their results add to our current understanding. For instance, are they providing an important update to the magnitude of the comparative differences, or greater clarity about relative impacts, or a more granular level of detail not available in previous studies but which are potentially important? For example, something I don't remember in other studies is an analysis like that shown in Figure 5b, which seems like an outcome that could be elevated and used as a basis for the broader discussion that incorporates consideration for (grid) energy transitions.

201 – 204, Figure 3, Figure 5,..: While it is interesting that the authors include Recycled scrap in the analysis, I would expect some additional context for how to interpret its relevance. For instance, what are the relative material flows of scrap vs recycled batteries in secondary supply chains (currently or projected for the future)? For example, if scrap-to-recycled battery stream ratio is 1:100, then the additional reduction in water consumption from 74% for battery streams to 88.4% for scrap does not really amount

to much, and is not particularly important to designing strategies for water efficiency and resilience in circular supply chains.

ADDITIONAL COMMENTS

Figure 1: What is the distinction between a conventional “material refinery facility” and a circular “recycling facility” in Figure 1b?

Figure 2 – Process descriptions: It would help if the authors included (perhaps as supplemental material) the details of the underlying process descriptions for the different pathways illustrated, specifically to help with distinguishing between Redwood process and “Representative Circular (RC)” pathway. For instance, do they share common/similar process steps and are the assumptions about these overlapping steps identical? Do they use the same chemicals? What degree of chemicals/solvent recovery or reuse is assumed? Does the RC Hy* process (in Figure 3) also include mechanical processing - which one would expect, but which is not obvious from the figure?

Figure 3 caption: Please correct the **b** and **c** captions of Figure 3 – it appears they are switched.

Lines 234 – 249: It is good to see that the authors highlighted these differences in predicted LCA impacts between studies, even if just for reference. However, how exactly should we interpret the results from the current work against the backdrop of those other results? Are there obvious limitations in other studies that is corrected for here? Can we have greater transparency in the underlying process and data for the current work?

Lines 258 – 260: The authors suggest that in conventional supply chains, producing mixed salts has higher energy and carbon footprint compared to producing discrete salts. This result is counter-intuitive, since I'd expect that producing individual salts would theoretically include an additional separation step, which adds energy, and consequently, CO₂ emissions. Could the authors explain this non-intuitive trend, and how come it is different from that seen for the circular pathway? It would help if they also provide the underlying process description as well.

Figure 6: I am not sure why Figure 6 is presented as a result, and not as supplemental information illustrating the approach used to model the transport logistics.

Lines 354 – 355, Figure 6: I think there is a very important element that is missing, **which is scale.**

- To compare primary and secondary logistics fairly, it would be good to try to make the scale more comparable. For example, expanding the collection radius from just California to say, the entire U.S. significantly changes the associated impact footprints. While I expect this to still be smaller than conventional, the comparison at this scale will be more justifiable.
- The converse argument is that for a much smaller scale of primary manufacturing, it is possible to identify a domestic supply chain that ends up with a much smaller footprint than the transnational supply chain considered in the model in Figure 6.

Figure 7: Can the authors clarify the reason for the difference in transport energy and CO2 footprint between EV and smartphone batteries? Does the difference arise from relative battery compositions, or transport distances? In addition, are there any key differences in their respective recycling steps, and how do these translate to differences in overall lifecycle impacts?

Lines 485 – 487 – Can the authors say more/ be a bit more specific here? Anyone can mention this trade-off without analysis. I would expect that the current analysis can supply enough information to recommend which of the two options – disassembly before transport vs transport before disassembly - is potentially a more impactful approach?

Lines 489 – 490: A number of studies mention the potential of rail transport to reduce energy and environmental impacts of secondary battery supply chain logistics relative to trucking, but how practical is this, especially for this analysis which is restricted to California? I imagine there are major limitations that make this nearly impossible, such as inadequate local rail coverage, competition with other high value cargo, etc.

Impact allocation method: Could the authors clarify what type of impact allocation method they used. For instance, was it based on the embodied economic value of the target products in the recycled or primary pathways?

Responses to Reviewers

Manuscript Title: Life cycle comparison of industrial-scale lithium-ion battery recycling and mining supply chains

Manuscript ID: NCOMMS-23-46463

In this document, we respond point-by-point to comments from the reviewers and the editor on our original manuscript submission. We thank the reviewers for their insightful feedback, and have incorporated several of their suggestions to further improve the manuscript before publication. The legend for this document is as follows:

Reviewer comments are in italics.

Author responses are in standard type and indented.

Deletions and additions to the original manuscript are marked in red and blue texts, respectively.

Line numbers are specified for either the original manuscript or the revised manuscript.

■ REVIEWER 2

Reviewer: *This study reveals that recycling LIBs into battery-grade materials significantly reduces environmental impacts. The research highlights that electricity consumption is the major factor in the environmental impacts of LIB recycling. It emphasizes the relatively minor contribution of material extraction and transport in circular supply chains, while these steps have a more notable impact in traditional supply chains. The inclusion of primary industrial-scale data from Redwood Materials is an interesting contribution, adding some real-world relevance to the findings. However, some arguments may not be fully supported, and some descriptions of important points are inadequate. The following are the detailed comments and critiques to be addressed:*

Our response: We are grateful to the reviewer for the favorable evaluation. Below, we have addressed the constructive comments with a focus on improved descriptions and support for our major arguments based on practical data from an operational battery recycling facility.

Reviewer: *1. While the study mentions the potential to help recyclers optimize environmental impacts, it would be strengthened by providing more concrete examples or strategies for such optimization. Having Redwood Materials as a case study is interesting, but there are several others using more mature or advanced facilities in US or global markets.*

Our response: We thank the reviewer for the suggestions, and take this opportunity to highlight our key insights related to environmental impacts of the lithium-ion battery supply chains. Notably, our study does provide concrete suggestions for optimizing environmental impacts in battery refinement. In addition, our analysis includes a case study, but has generalizable contributions to the literature enabled by granular Redwood data and our rigorous analysis of gate-to-gate and cradle-to-gate environmental impacts. Our response below highlights two major areas: (1) insights

and strategies for optimizing environmental impacts and (2) importance and broader impacts of employing data from Redwood Materials.

1. *Insights and strategies for optimizing environmental impacts.*

The first novelty of our analysis is that it provides quantitative evaluations for industrial-scale circular refinement of different feedstocks using different technologies within the same process flow. While both recycled scrap and batteries are projected to be important recycling sources in the coming decades, our findings quantify the different environmental impacts of refining each feedstock, and give concrete suggestions for advancing corresponding pyrometallurgical and hydrometallurgical technologies. We have revised the manuscript to strengthen these novel contributions (lines 262–267, 524–533, 541–546):

“Similar to RC, emerging oxygen-free pyrometallurgical processes rely on carbothermic and thermite reduction for recycling cathode metal compounds at moderate temperatures (600–1000°C)^{19,21}. Our results show that the RC step accounts for 5.5%–7.4% of the total environmental impacts in Redwood (Fig. 3), demonstrating the environmental feasibility of industrial-scale carbothermal reduction-based pyrometallurgy.”

“Our findings showed the promise of carbothermal reduction-based pyrometallurgy for pretreatment of LIB recycling; however, we note that controlling process conditions to achieve optimal environmental impacts and product formats will elevate the costs. Existing hydrometallurgy exhibits advantages over pyrometallurgy in terms of energy efficiency and refinement precision, but requires costly pretreatment and purification, and suffers from limited scalability and from generation of secondary waste liquids^{34,35}. ~~Because~~ Our analysis identified chemical consumables such as H₂O₂ ~~are~~ as important contributors to hydrometallurgy, indicating that environmental impacts of Hy processes ~~could~~ can be reduced through more sustainable (e.g., electrochemical)–production methods for chemical inputs (e.g., electrochemical generation of H₂O₂)³⁶.”

“Our findings provide important guidance for sources and products in future refinement. Battery scrap is currently the primary recycling source due to the proliferation of gigafactories, but will decrease in the future with quality control advances in manufacturing⁸. While refining recycled batteries required higher greenhouse gas emissions, energy input, and water demand than refining battery scrap (Fig. 3), it is critical to improve the technologies for recycling energized batteries for future challenges.”

Second, by contrasting the environmental impacts of producing mixed metal salts and discrete salts (Fig. 4), we demonstrated the environmental and economic benefits of refining batteries to mixed metal salts, while most current recycling facilities are still producing discrete salts. We highlighted such optimization strategies in lines 318–321 and 547–549.

“Overall, our results indicated that refining batteries to mixed metal salts instead of discrete salts can substantially save environmental impacts while still satisfying the needs of circular LIB supply chains. Our findings also provide important insights to optimizing plant-scale battery refining operations.”

“Our findings also demonstrated the benefits of mixed metal sulfates as refinement products over single salts in the circular supply chain, indicating that the further separations between Ni and Co salts can be avoided.”

Third, we quantified the contributors to environmental impacts in various circular refinement technologies compared side-by-side at the same facility (**Fig. 5**), providing suggestions for optimizing the circular environmental impacts using suitable input electricity. These suggestions are informed by our unique focus on practical facility-level data regarding overall and element-specific mass balances, along with key inputs such as energy, water, and chemical reagents. In contrast, many existing studies refer to other reports for primary operational data and do not benefit from the iterative insights enabled by our academic-industrial partnership.

As an example of enacting insights from our life cycle assessment, we note that Redwood Materials *has already* referred to our findings and optimized their refining products accordingly, including (1) choosing mixed (Ni,Co)SO₄ as their major refining products to avoid unnecessary separations, (2) substituting the original Nevada Power Company-based input electricity with that generated from sustainable sources to lower environmental impacts, and (3) using key chemical inputs (e.g., H₂O₂) produced from low environmental-impact methods identified by our study. Additionally, our findings and suggestions have gained great interest from numerous battery production and battery recycling companies in conferences and industrial meetings. For example, the tradeoff we identified between water consumption and greenhouse gas emissions has been noted by several companies as a novel insight worth considering in the placement and design of pilot scale battery recycling facilities (lines 556–558):

“Therefore, considering water consumption and CO₂-eq emissions is necessary for selecting recycling facility locations, particularly in water-sensitive or emissions-sensitive scenarios.”

2. Importance and broader impacts of employing data from Redwood Materials.

Our analysis based on technologies and data from Redwood Materials is not only a specific *case study*; rather, utilizing the state-of-the-art and versatile technologies and practical data in Redwood enable us to investigate different factors in the circular LIB refinement from a more granular level than commonly provided in similar literature reports, and provide important insights for the whole area of LIB recycling.

First, Redwood Materials represents state-of-the-art circular refinement technologies that are comparable to existing and emerging technologies. Reductive calcination (RC) is a pyrometallurgical process relying on carbothermal reduction with optimized conditions. Pyrometallurgy based on carbothermal reduction has been rapidly developed at lab scale. Our study, to the best of our knowledge, reports the first evaluation of the industrial-scale application of the emerging technology, providing important insights to all pyrometallurgical refinement technologies. The hydrometallurgical processes at Redwood that refine both energized batteries and scrap are analogous to the prevailing hydrometallurgy, and our analysis identifying environmental impact contributors to the Hy process similarly advances Hy processes beyond Redwood. The manuscript has been revised to better generalize the working conditions of Redwood technologies, and compare with existing refinement methods (lines 513–528):

“Disaggregated analysis of LIB refinement pathways at Redwood Materials ~~provides~~ provided important insights into the performance and potential of different refinement processes. Pyrometallurgical technologies are advantageous in scalability and operating safety^{9,10,33}, ~~While pyrometallurgical processing is~~ but are widely considered as ~~more~~ environmentally intensive ~~than hydrometallurgy^{31,32}~~ due to the high reaction temperature⁷. Oxygen-free carbothermic reduction and thermite reduction have been recently investigated at lab scale for energy-efficient LIB cathode

pyrolysis¹⁹⁻²¹. Our analysis showed that Redwood Materials' industrial-scale carbothermal reduction-based RC-pathway pyrometallurgy (reductive calcination in Redwood Materials) exhibits exhibited much lower environmental impacts than prevailing direct roasting and smelting pyrometallurgical and current Hy-containing hydrometallurgical pathways reported in practice and in literature (Fig. S3). The optimized conditions of RC processing minimizes the combustion of carbon-containing LIB materials, significantly reducing CO₂-eq emissions while simultaneously generating products that are amenable for hydrometallurgical separation. Our findings showed the promise of carbothermal reduction-based pyrometallurgy for pretreatment of LIB recycling; however, we note that controlling the process conditions to achieve optimal environmental impacts and product formats will elevate the costs.

Second, operational data from Redwood Materials provide more granular insights to verify and correct the existing model-based studies. This Redwood-based study helped identify the gaps between models and industrial operations, and we highlighted the needs for advancing the whole field in the revised manuscript (lines 531–541):

“~~Because~~ Our analysis identified chemical consumables such as H₂O₂ ~~are~~ as important contributors to hydrometallurgy, indicating that environmental impacts of Hy processes ~~could~~ can be reduced through more sustainable ~~(e.g., electrochemical)~~ production methods (e.g., electrochemical generation of H₂O₂)³⁶.”

“We note the inconsistencies among existing studies on assessing conventional LIB refinement due to different product compositions, which underscores the need to standardize the functional unit in reporting environmental analyses for future studies in the field. The deviations between our results of circular refinement and model-based literature data highlight the importance of reconciling models with industrial operating data.”

Third, the versatile technologies at Redwood Materials enable treatment of different feedstocks (energized batteries and battery scrap) and generation of different product formats (mixed metal sulfate and discrete salts), offering opportunities to investigate influences of multiple factors on the circular refinement. Using Redwood data, we analyzed the refinement of energized batteries and scrap, providing suggestions for developing generalizable strategies for different recycling sources (Fig. 3 and lines 542–547). Furthermore, while prevailing recycling produces discrete salts, we analyzed different refinement scenarios and demonstrated the advantages of producing mixed salts (Fig. 4 and lines 315–320).

“Our findings provided important guidance for sources and products in future refinement. Battery scrap is currently the primary recycling source due to the proliferation of gigafactories, but will decrease in the future with quality control advances in manufacturing⁸. While refining recycled batteries requires higher greenhouse gas emissions, energy input, and water demand than refining battery scrap (Fig. 3), it is critical to improve the technologies for recycling energized batteries for future challenges.”

“The RC pathway (RC+Me+Hy) ~~exhibits~~ exhibited lower energy consumption (–72.3%), CO₂-eq emissions (–39.5%), and water consumption (–12%) relative to the Me+Hy pathway (Fig. 4), because it avoids additional treatment separating (Ni,Co)SO₄ to discrete salts. Overall, our results ~~indicate~~ indicated that refining batteries to mixed metal salts instead of discrete salts can substantially save environmental impacts while still satisfying the needs of circular LIB supply chains.”

In addition to Redwood, we are aware of other industrial-scale facilities for battery recycling, but the complete operating data are not public or cannot be readily accessed. Our analyses relying on practical data from Redwood provide exceptionally important and granular insights to several research communities (e.g., battery recycling, life cycle assessment, battery synthesis), and can spark more academic-industrial collaborations to further advance the field.

The manuscript has been revised to better highlight our contributions (lines 147–152):

“With the methodologies and results reported in this study, researchers can prioritize major opportunities to improve process efficiencies, practitioners can benchmark their environmental impacts, and policymakers can incentivize best environmental practices in LIB supply chain management. Granular insights provided by this study based on practical data can also help recyclers optimize the environmental impacts of their refinement processes, and spark more academic-industrial collaborations to further advance the field.”

Reviewer: 2. *The study appears to be well-structured and clear in its objectives and methodologies. However, ensuring that the data sources, assumptions, and limitations are transparently communicated will be crucial for the study's credibility and usefulness. More importantly, the data used in Fig 1 is based on market value from 2018 to 2021, which is quite outdated. In the battery industry, the prices change in weekly or even daily basis, and fluctuation can be large. One typical exam is Li, its prices have changed by about 75% since 2022.*

Our response: We agree with the reviewer regarding the importance of transparent communication regarding data sources, assumptions, and limitations. To this end, we have included a robust Limitations section in the manuscript, and have cited all data sources appropriately for reproducible analysis. We were especially careful regarding reproducibility and transparency because not all existing reports include data that can achieve these goals. Our commitment to reproducible analysis shows the rigor and contributions to the field of the submitted study. Regarding Figure 1, the purpose is not to reflect the most updated pricing, but to highlight the motivations; with this purpose in mind, our data sources limitations are very transparent. We are well aware of the rapidly changing market prices of common recycled commodities and lithium-ion batteries, but the purpose of presenting Fig. 1a is for highlighting the important *drivers* for recycling LIBs. Of note, these prices reported are spot market prices, and are meant to show trends across critical materials and battery chemistries. We chose the years 2018–2021 because that 2021 is when our study started and reflects the timing of the original data from Redwood Materials as well.

Throughout the manuscript, except for the confidential information from our industrial collaborators, we have provided all the data sources in Supplementary Information, clearly stated the assumptions in the main manuscript, and summarized the study limitations in the Methods section.

Reviewer: 3. *The methodology seems robust, utilizing the GREET 2021 model for conventional supply chain analysis and incorporating data from actual industrial operations for the circular supply chain. However, the model is also quite outdated.*

Our response: We thank the reviewer for raising the issue and have carefully examined the updates in GREET 2023. Compared with the 2021 version, GREET 2023 provided additional refining pathways for Ni and Al and updated the environmental impacts for several relevant components. Employing data from GREET 2023 yields 17.0%–18.3% and 2.2%–7.4% higher environmental impacts than those in the submitted manuscript for producing discrete and mixed salt, respectively (**Figs. 3–4**). No change has been identified for the life cycle inventory of consumables, i.e., the environmental impacts of circular refinement based on Redwood data (**Figs. 3–5**) are not influenced by updating GREET. Overall, applying GREET 2023 data will increase the savings of environmental impacts for circular compared to conventional refinement, but will *not* change the main conclusions of the study.

First shaped in 2021, the study analyzed the practical data from Redwood based on their operations in 2021, and employed the GREET 2021 model to guarantee a fair comparison with the conventional supply chains. We keep the results based on GREET 2021 in the main manuscript with further data corrections, and have provided an additional **Supplementary Data File B** appended to the revised manuscript with results based on GREET 2023 to compensate for the limitations of the 2021-based analysis. This approach prioritizes similar time periods for the conventional data for GREET and the circular data from Redwood; it also conservatively estimates (i.e., underestimates) savings provided by circular compared to conventional refinement approaches (lines 191–197).

*“Excluding the environmental impacts of material extraction and transport steps, the gate-to-gate production of one kg NCA-eq battery-grade material from state-of-the-art conventional mined natural materials ~~consumes~~ consumed 193.9 MJ and 77.3 L H₂O while emitting 14.5 kg CO₂-eq (**Fig. 3**). Note that the results are based on GREET 2021 to match the period when circular refinement data were collected at Redwood; more recent environmental impacts of the conventional supply chain based on the 2023 model are 17.0%–18.3% higher than that of 2021, and are presented in Supplementary Data File B.”*

Reviewer: 4. *The results indeed are different from some of the recent findings that transportation can be a significant cost factor, especially collecting spent lithium-ion batteries and transport them as Class 9 hazardous wastes is a huge cost burden. But the authors seem to have not included this factor in discussing recycling as those spent batteries will not automatically concentrate in the recycling facility.*

Our response: We are aware of the different understanding of transport costs in LIB recycling, but note that the current study is focused on *environmental impacts*, with emphasis on the *refinement* step. Examining cost considerations in the circular battery supply chain warrants additional investigation and is beyond the scope of the current study.

The manuscript has been revised to reflect the importance of these considerations for future research (lines 600–603, and 606–610):

“Business models for collection of all LIB types and sizes will likely vary from manufacturer-led to municipal or private collection programs., and can be influenced by the safety costs while collecting used batteries as hazardous wastes.”

“Additionally, designing and manufacturing LIBs for recycling in a circular economy can reduce resource usage identified in this study⁴². Recyclable battery packing can save energy and chemical

inputs used in refinement, and manufacturing battery cells and modules favorable for extraction and integration will lower the environmental impacts of the upstream steps.”

Reviewer: 5. The analysis comparing pyrometallurgical and hydrometallurgical processing, and the identification of the environmental benefits of the RC pathway, is valuable. However, the study could benefit from a deeper exploration of the trade-offs between these methods, particularly in terms of scalability and economic viability.

Our response: We are grateful to the reviewer for the helpful suggestion. Lines 267–270 in the original manuscript compare tradeoffs between different methods:

“Note that traditional pyrometallurgy and Redwood Material’s reductive calcination can process energized batteries of varying states of charge, health, and formats with minimal modification, whereas traditional hydrometallurgy may need to discharge energized batteries in salt bath or cryogenically remove electrolyte for safe mechanical processing.”

We also added lines 513–528 and 528–534 in the revised manuscript to extend the discussion as suggested:

“Disaggregated analysis of LIB refinement pathways at Redwood Materials ~~provides~~ provided important insights into the performance and potential of different refinement processes. Pyrometallurgical technologies are advantageous in scalability and operating safety^{9,10,33}. ~~While pyrometallurgical processing is~~ but are widely considered as ~~more~~ environmentally intensive ~~than hydrometallurgy~~^{31,32} due to the high reaction temperature. Oxygen-free carbothermic reduction and thermite reduction have been recently investigated in lab scale for energy-efficient LIB cathode pyrolysis³³⁻³⁵. Our analysis showed that Redwood Materials’ industrial-scale carbothermal reduction-based RC pathway pyrometallurgy (reductive calcination in Redwood Materials) exhibits exhibited much lower environmental impacts than prevailing direct roasting and smelting pyrometallurgical and current Hy-containing hydrometallurgical pathways reported in practice and in literature (Fig. S3). The optimized conditions of RC processing minimizes the combustion of carbon-containing LIB materials, significantly reducing CO₂-eq emissions while simultaneously generating products that are amenable for hydrometallurgical separation. Our findings showed the promise of carbothermal reduction-based pyrometallurgy for pretreatment of LIB recycling; however, we note that controlling the process conditions to achieve optimal environmental impacts and product formats will elevate the costs.”

“Existing hydrometallurgy exhibits advantages over pyrometallurgy in terms of energy efficiency and refinement precision, but requires costly pretreatment and purification, and suffers from the limited scalability and generation of secondary waste liquids^{34,35}. ~~Because~~ Our analysis identified chemical consumables such as H₂O₂ ~~are~~ as important contributors to hydrometallurgy, indicating that environmental impacts of Hy processes ~~could~~ can be reduced through more sustainable (e.g., ~~electrochemical~~) production methods (e.g., electrochemical generation of H₂O₂)³⁶.”

Reviewer: 6. The study's focus on upstream impacts, including extraction and transport, is an important aspect that often gets overlooked. The suggestion for further concentrating materials near mine sites or adjusting locations of refineries to reduce environmental impacts is a practical recommendation.

Our response: We thank the reviewer for the favorable comments on our findings and their novel contributions to the literature.

Reviewer: 7. The mention of designing and manufacturing LIBs for recyclability is a crucial point, suggesting a shift towards a more sustainable, circular economy. However, this aspect could be explored in more detail, particularly in terms of how current manufacturing practices could be adapted.

Our response: The manuscript has been revised to extend the discussion (lines 606–610):

“Additionally, designing and manufacturing LIBs for recycling in a circular economy can reduce resource usage identified in this study³⁰. Recyclable battery packing can save energy and chemicals used in refinement, and manufacturing battery cells and modules favorable for extraction and integration will lower the environmental impacts of the upstream steps.”

Reviewer: 8. Give a clearer, more detailed explanation of methodologies, especially where the authors mention modeling and the “shortest route” approach. This provides context and helps readers understand how your conclusions were reached.

Our response: We have provided detailed descriptions in the submitted manuscript of the modeling of extraction and transport in **Note 4** and **Note 5** in **Supplementary Information**, respectively. Representative descriptions are listed below:

*“California (CA) was chosen because it is the most populated US state (39.6 million, 2022) and has the highest number of electric vehicles, representing 42% of the entire country in 2020¹⁸. Existing Collection Facilities (CFs) are provided by the California Department of Resources Recycling and Recovery (CalRecycle)¹⁸. Two different LIB products are considered: end-of-life LCO-based smartphones and end-of-life NCA-based EV batteries. The assumed mass of transported smartphones and EV battery packs along with the mass of active cathode material is summarized in **Table S10**.”*

*“In the first scenario, transport of smartphones from the census-block level to the nearest CF is used to estimate environmental impacts of extraction from consumers. Population in every census block is obtained from the U.S. Census Bureau¹⁹. It is assumed that one smartphone is owned by each person in CA and is recycled every three years. To determine the closest CF, k-means clustering is used for the allocation of census blocks to CFs based on minimizing the distance between a census block’s centroid and a singular CF, for all census blocks. This process produces clusters of block groups whose LIB-embedded smartphones are allocated to the closest CF. The distance from the census blocks to the CFs is determined by the shortest route using Dijkstra’s algorithm. The number of clusters, “k,” used in this algorithm is fixed and equals the total number of CFs. The relative size of a CF in the inset of **Fig. 6** indicates the number of collected smartphones as compared to nearby CF clusters, and the smartphone collection area is color-coded for distinction for each CF. Multiplying the resulting distances with the average mass of a smartphone yields a mass-distance product (i.e., kg × km), which are converted to environmental impacts using values for trucking (**Table S12**). Note that the shortest-route value represents a lower bound of emissions.”*

“For a particular element, the shortest route utilizing all combinations between mines and refineries is calculated with Dijkstra’s algorithm²⁷.”

*“The normalized mass-based environmental impacts of LIB material concentrate transported, \bar{I}_i , are calculated by Eq. S, where I_i is mass-based environmental impacts of the material i with a certain transport model (**Table S9**), d_i is the segmental transport distance in the shortest-route model, n_d*

is the total segments along that route, and w_i is the element weight percent from the country of origin.

$$\bar{e}_{\text{Mat},i}^m = \frac{\sum_{i=1}^{n_d} (e_{\text{Mat},i}^m d_i)}{w_i} \quad (S1)$$

Environmental impacts with weighted distributions, $E_{\text{Mat},i}$, for each element i are calculated to consider all routes for the three environmental impacts:

$$E_{\text{Mat},i} = \sum_{i=1}^{n_d} (\bar{e}_{\text{Mat},i}^m m_i) \quad (S2)$$

where m_i denotes the global percentage of an element considered for mined production or refinement production in 2019 (values summarized in **Tables S9–S10**)."

We have added lines 407–413, 446–451, 722–727, 731–734 in the main manuscript and Methods to better summarize the methodology:

~~“Smartphones ~~are~~ were considered extracted when collected, aggregated, and transported from all California residents (analyzed per census block) to the nearest existing collection facility (CF)²⁴. ~~The analytical model for this circular extraction is depicted in Fig. 6a, where a Δ shortest-path route for collection ~~is~~ was determined by minimizing the distances from block group to CF ~~is modeled~~ for the whole state (the model is summarized in **Methods** and detailed in **Note 4 in Supplementary Information**)³⁰.~~~~

~~“An algorithm is developed to quantify The environmental impacts of transport ~~are~~ were quantified by based on a weighted distribution of participating countries and calculating the shortest distance along major transport routes among the participating countries weighted by the relative contributions of the countries to the market (the case of cobalt is presented as an example in **Fig. 6b**). Details of the modeling method are presented in Note 5 of Supplementary Information.”~~

~~“A shortest-route ~~algorithm~~ method was ~~used~~ developed for modeling LCO collection at the closest municipal collection facility. Census blocks centered by their CFs were first determined for the state of California by k-means clustering method, and the pathways of transporting LCOs to CFs were identified through minimizing the possible travel distance, realized by the Dijkstra’s algorithm⁵⁰ (details of the methods are described in **Note 4 in Supplementary Information**).”~~

~~“The distances of the shortest-path routes ~~are~~ were calculated between mines and refineries by country, predicated on the closest available modes of transport (including road, rail, and maritime). Details of the modeling method are described in Note 5 of Supplementary Information.”~~

Reviewer: 9. *Where discussing environmental impacts, also consider mentioning the economic aspects more explicitly. Balancing environmental benefits with economic feasibility is crucial for practical applications.*

Our response: We acknowledge the importance of balancing environmental and economic considerations, but emphasize that the current study is focused on *environmental impacts*, especially of the refinement step; economic feasibility is beyond the scope of the current study and warrants further investigation. Thus, we have restricted our claims to the environmental analysis conducted. We have mentioned this scoping point in the original main manuscript (lines 604–606, and 610–612):

“Notably, analogous economic and environmental impacts to local ecosystems of conventional mining are not considered in this analysis, and warrant future studies⁴¹.

“Future efforts should also focus on optimizing refinement processes for subsequent steps of the circular supply chain in LIB manufacturing, product performance, and economic cost.”

We have also added lines 519–528 and 528–534 to mention the economic aspects in the discussion of the present results (bolded text refers explicitly to costs):

*“~~Our analysis showed that Redwood Materials’ industrial-scale carbothermal reduction-based RC pathway pyrometallurgy (reductive calcination in Redwood Materials) exhibits~~ **exhibited** much lower environmental impacts than ~~prevailing direct roasting and smelting pyrometallurgical and current Hy-containing hydrometallurgical pathways reported in practice and in literature~~ (Fig. S3). ~~The optimized conditions of RC processing minimizes the combustion of carbon-containing LIB materials, significantly reducing CO₂-eq emissions while simultaneously generating products that are amenable for hydrometallurgical separation.~~ **Our findings showed the promise of carbothermal reduction-based pyrometallurgy for pretreatment of LIB recycling; however, we note that controlling the process conditions to achieve optimal environmental impacts and product formats will elevate the costs.**”*

*“~~Existing hydrometallurgy exhibits advantages over pyrometallurgy in terms of energy efficiency and refinement precision, but requires costly pretreatment and purification, and suffers from limited scalability and from generation of secondary waste liquids^{34,35}. Because~~ **Our analysis identifies** chemical consumables such as H₂O₂ ~~are as~~ **are as** important contributors to hydrometallurgy, ~~indicating that~~ **environmental impacts of Hy processes could can** be reduced through more sustainable ~~(e.g., electrochemical)~~ **production methods for chemical inputs (e.g., electrochemical generation of H₂O₂)³⁶.**”*

Reviewer: 10. *The study's broader implications for strategic placement of LIB facilities and policy recommendations are valuable. However, more concrete suggestions or models for policy implementation would enhance the study's practical utility. It would also be beneficial if the study included specific recommendations or identified key areas where policy interventions could be most effective.*

Our response: We thank the reviewer for the favorable comments on the broader implications of our work, but politely note that the current study is focused on *technically* investigating the environmental impacts of battery recycling; proposing specific recommendations on policy intervention is beyond the scope of the current study. Instead, we anticipate that future policy-focused studies and interventions will leverage our results to inform policy analysis. However, we have highlighted the potential implications of our findings for advancing the field in terms of optimization and standardization, which are closely related to policy intervention (lines 556–558):

“Therefore, considering water consumption and CO₂-eq emissions is necessary for selecting recycling facility locations, particularly in water-sensitive or emissions-sensitive scenarios.”

We also added lines 531–541 and 606–610 to the revised manuscript to strengthen the discussion:

*“~~Because~~ **Our analysis identified** chemical consumables such as H₂O₂ ~~are as~~ **are as** important contributors to hydrometallurgy, ~~indicating that~~ **environmental impacts of Hy processes could can** be reduced through more sustainable ~~(e.g., electrochemical)~~ **production methods (e.g., electrochemical generation of H₂O₂)³⁸.** ~~...We note the inconsistencies among existing studies on~~*

assessing conventional LIB refinement due to different product compositions, which underscores the need to standardize the functional unit in reporting environmental analyses for future studies in the field. The deviations between our results of circular refinement and model-based literature data highlight the importance of reconciling models with industrial operating data.”

“Additionally, designing and manufacturing LIBs for recycling in a circular economy can reduce resource usage identified in this study⁴². Recyclable battery packing can save energy and chemicals used in refinement, and manufacturing battery cells and modules favorable for extraction and integration will lower the environmental impacts of the upstream steps.”

Reviewer: 11. Expand on the limitations of study and the implications for future research. This not only adds credibility to your work but also guides future studies in this area.

Our response: We are grateful to the reviewer for the suggestion. We note that a subsection **Summary of Study Limitations in Methods** is provided in **Methods**, and has been expanded in the revised manuscript. Specifically, lines 531–541 and 600–603 have been added to the revised manuscript to strengthen the implications:

~~“Because—Our analysis identified~~ chemical consumables such as H₂O₂ ~~are as~~ important contributors to hydrometallurgy, indicating that environmental impacts of Hy processes ~~could can~~ be reduced through more sustainable ~~(e.g., electrochemical)~~ production methods (e.g., electrochemical generation of H₂O₂)³⁸.~~...We note the inconsistencies among existing studies on assessing conventional LIB refinement due to different product compositions, which underscores the need to standardize the functional unit in reporting environmental analyses for future studies in the field. The deviations between our results of circular refinement and model-based literature data highlight the importance of reconciling models with industrial operating data.~~”

“Business models for collection of all LIB types and sizes will likely vary from manufacturer-led to municipal or private collection programs, and can be influenced by the safety costs while collecting used batteries as hazardous wastes.”

Reviewer: Minor comments:

1. Some phrases might be ambiguous or unclear. For example, line 478: "building reinterests closer to sources" is unclear and might need rephrasing based on the intended meaning.

Our response: We have revised the typographical errors throughout the manuscript, including in lines 479–481 and 566–568:

“Circular production of LCO-grade materials leads to higher environmental impacts than that of NCA-grade materials based on the mixed-stream feedstock composition processed by Redwood Materials.”

“Further concentrating materials near mine sites or building ~~reinterests~~ refineries closer to sources can efficiently reduce environmental impacts of the conventional mined materials.”

Reviewer: 2. Maintain a consistent verb tense throughout the text. For example, switch from "The analysis reveals" to "The analysis revealed" if the rest of your text is in past tense.

Our response: We thank the reviewer for noticing the differences in tense, and have adjusted actions taken during our study to past tense. Some example changes are presented below (lines 176–180, 292–297, 332–338, 401–407, and 467–472):

“Refining lithium-ion batteries into battery-grade materials ~~exhibits~~ exhibited lower environmental impacts than production from mined natural materials. Environmental impacts of refinement pathways in conventional and circular LIB supply chains were analyzed in Fig. 3, while the ~~The~~ upstream steps of material extraction and transport ~~are~~ were considered in later sections.”

“Formats of refinement products ~~influence~~ influenced environmental impacts. ...To examine the influences of the refinement product formats, the environmental impacts of refinement to mixed salt ~~are~~ were compared to the refinement to discrete sulfate salts, NiSO₄ and CoSO₄ (Fig. 4). Both conventional and circular refinement pathways ~~are~~ were analyzed.”

“Electricity consumption ~~dominates~~ dominated the environmental impacts of lithium-ion battery circular refinement. The relative environmental impacts of input consumables (e.g., energy, water, commodity chemicals) in the gate-to-gate refinement processes ~~are~~ were disaggregated in Fig. 5 (additional criteria air pollutants in Tables S2–S3, Figs. S4–S5). Note that the embodied environmental impacts of electricity consumption in Fig. 3 ~~are~~ were based on the Nevada Power Company (NEVP) at the Redwood Materials location. Electricity consumption ~~is~~ was found to be a principal factor dominating the environmental impacts.”

“Environmental impacts of material extraction and transport ~~are~~ were significantly lower in circular lithium-ion battery supply chains than in conventional supply chains. ...Environmental impacts of these upstream steps ~~are~~ were analyzed for two representative LIB chemistries and battery use cases: NCA in EV battery packs, and lithium cobalt oxide (LiCoO₂ or LCO) in smartphones. California ~~is~~ was chosen to assess circular extraction because...”

“The refinement step ~~dominates~~ dominated environmental impacts of circular and conventional supply chains. Combining material extraction, transport, and refinement steps ~~yields~~ yielded a cradle-to-gate comparison...”

Reviewer: 3. Thoroughly proofread the text to catch any typographical errors, misplaced commas, or incorrect word usage.

Our response: We thank the reviewer for the close read; we have thoroughly proofread the document and made several improvements for readability, including (lines 94–96):

“The most significant environmental differences between LIB production from circular and conventionally mined cathode material lie early in supply chains, comprised of extraction, transport, and refinement steps (together as “cradle-to-gate,” Fig. 1b).”

Additional changes were made (including the previous comments directly above), but are enumerated in detail here for brevity.

Reviewer: 4. Where you refer to figures (e.g., “Fig. S3”, “Fig. 5”), ensure these are easily accessible to the reader and consider including brief descriptions in the text to aid understanding for those who might not have immediate access to the figures.

Our response: We note that *Nature Communications* is an open access journal where the readers can access the main manuscript and source data including the supplementary figures. We have

provided adequate descriptions in the original manuscript while citing the figures, e.g. (lines 155–156, 228–229, 333–335):

“In LIB supply chains, the refinement step converts the collected feedstocks into battery-grade salts for further manufacturing (Fig. 2a).”

“The environmental impacts of two circular refinement pathways ~~are~~ were presented in each graph in Fig. 3 for mixed-stream LIB feedstocks processed at Redwood Materials...”

“The relative environmental impacts of input consumables (e.g., energy, water, commodity chemicals) in the gate-to-gate refinement processes ~~are~~ were disaggregated in Fig. 5 (additional criteria air pollutants in Tables S2–S3, Figs. S4–S5).”

Additional figure descriptions (lines 197–199, 271–274, 393–395, and 443–446) were added to the revised manuscript to improve clarity:

“The values of ~~energy consumption and~~ greenhouse gas emissions ~~are~~ were comparable with previous studies based on GREET datasets^{12,15} (comparison of environmental impacts with data in literature is detailed in Fig. S3b).”

“While this analysis ~~is~~ was focused on Redwood Materials refinement pathways, the methodology can be used to evaluate additional refinement pathways (e.g., analysis of the representative hydrometallurgy was detailed in Fig. S3c), or others that use different material feedstocks, refinement processes, and energy supplies.”

This tradeoff also ~~explains~~ explained the different influences of electricity source on environmental impacts of the Redwood Materials refinement step and other pathways (detailed in Fig. S3d).

After extraction, LIB material concentrates are transported along domestic and international routes by truck, train rail, and maritime cargo ship to refinery locations (a portion of the network model is presented in Fig. S76, and data are summarized in Table S8–S15; complete references in Supplementary Information).

Reviewer: 5. Many references are in random formats.

Our response: We thank the reviewer for raising the errors, and have adjusted the citations below. Note that the corrected reference numbers may change in the revised manuscript. Some original references had missing information, including Refs. 2–3, 22–24, 27, 29, 33, and 36–37. We have revised the respective citations to:

“2 National blueprint for lithium batteries 2021–2030. (Department of Energy, 2021).”

“3 Global supply chains of EV batteries. (International Energy Agency, 2022).”

“30 Electric Vehicle Registrations by State. (U.S. Department of Energy, 2022).”

“31 United States Census Bureau Database. (United States Census Bureau, 2020).”

“32 Where to recycle: Map of public recycling locations. (CalRecycle, 2022).”

“39 Hazardous Materials: Transportation of Lithium Batteries. (Pipeline and Hazardous Materials Safety Administration, 2014).”

“41 Daily, G. C. & Ruckelshaus, M. 25 years of valuing ecosystems in decision-making. Nature 606, 465-466 (2022).”

“45 Korolev, I. et al. in 29th International Mineral Processing Congress, IMPC 2018.”

“47 Battery recycling through extraction of metals from spent lithium-ion batteries. (Syensqo, 2023).”

“48 *Creating a circular and sustainable battery supply chain. (Li-Cycle, 2023).*”

Other original references had incorrect journal abbreviations, including Refs. 4, 6–8, 11, 16–17, 25, 28, and 30. We have been revised to:

“4 *Winslow, K. M., Laux, S. J. & Townsend, T. G. A review on the growing concern and potential management strategies of waste lithium-ion batteries. Resour. Conserv. Recycl. 129, 263-277 (2018).*”

“6 *Mayyas, A., Moawad, K., Chadly, A. & Alhseinat, E. Can circular economy and cathode chemistry evolution stabilize the supply chain of Li-ion batteries? Extr. Ind. Soc. 14, 101253 (2023).*”

“7 *Jacoby, M. It’s time to get serious about recycling lithium-ion batteries. Chem. Eng. News 97, 29-32 (2019).*”

“11 *Ciez, R. E. & Whitacre, J. F. Examining different recycling processes for lithium-ion batteries. Nat. Sustain. 2, 148-156 (2019).*”

“12 *Crenna, E., Gauch, M., Widmer, R., Wäger, P. & Hischer, R. Towards more flexibility and transparency in life cycle inventories for Lithium-ion batteries. Resour. Conserv. Recycl. 170, 105619 (2021).*”

“24 *Makuza, B., Tian, Q., Guo, X., Chattopadhyay, K. & Yu, D. Pyrometallurgical options for recycling spent lithium-ion batteries: A comprehensive review. J. Power Sources 491, 229622 (2021).*”

“26 *Kim, S. et al. A comprehensive review on the pretreatment process in lithium-ion battery recycling. J. Clean. Prod. 294, 126329 (2021).*”

“37 *Roithner, C., Cencic, O. & Rechberger, H. Product design and recyclability: How statistical entropy can form a bridge between these concepts-A case study of a smartphone. J. Clean. Prod. 331, 129971 (2022).*”

“40 *Sun, M., Yang, X., Huisin, D., Wang, R. & Wang, Y. Consumer behavior and perspectives concerning spent household battery collection and recycling in China: a case study. J. Clean. Prod. 107, 775-785 (2015).*”

“42 *Berger, K., Schögl, J.-P. & Baumgartner, R. J. Digital battery passports to enable circular and sustainable value chains: Conceptualization and use cases. J. Clean. Prod. 353, 131492 (2022).*”

■ REVIEWER 3

Reviewer: *Review Summary: this paper presents a comparative life cycle analysis for primary and secondary supply chains for lithium ion batteries (LiBs). The paper is well written, the topic is of interest, the core methodology is sound, and outcomes relevant for advancing the conversation around designing circular supply chains for LiBs. The major concerns I have about this study have to do with the case for its uniqueness, the significance of some of the key outcomes and the appropriateness of the context for interpreting these results. I think it needs to be updated before considering it for publication. I provide these as comments below for consideration.*

Our response: We thank the reviewer for the favorable reception of our study. We have carefully considered their comments on uniqueness, significance, and results interpretation and revised the manuscript as detailed below.

Reviewer: *Lines 128 – 130, 438 – 440: The authors make the case that a distinguishing contribution from this study is the representation of (1) industrial scale circular refinement data, & (2) stepwise cradle-to-gate supply chain comparison, but it is not clear to me what this claim really amounts to.*

For (1), I imagine this has to do with using data from Redwood's process, so what is the value proposition then? Does using data from Redwood's process provide more accurate (or trustworthy) recycling impacts compared to previous studies? Since the authors correctly pointed out that there are significant differences in reported environmental impacts for LiB recycling (Lines 91-94), how does this study address and resolve that problem, and not simply add another data point to literature on LiB recycling impacts?

Our response: We thank the reviewer for raising the questions concerning the importance of analyzing Redwood Materials processes. In general, our analyses based on Redwood data **provide comprehensive and granular insights for advancing various refining technologies**, and Redwood processes **allow us to systematically investigate impacts of different factors in recycling LIBs**. Below we summarize the important value propositions and potential broad impacts of employing Redwood's processes, and have revised the main manuscript to better highlight these points. Note that this comment resembles the first comment from Reviewer 2; thus we have adapted our response from that comment (Page 1 of this document).

First, Redwood Materials represents state-of-the-art circular refinement technologies that are comparable to existing and emerging technologies, and our findings provide insights for advancing various circular refining technologies. Reductive calcination (RC) is a pyrometallurgical process relying on the carbothermal reduction with optimized conditions. Pyrometallurgy based on carbothermal reduction has been rapidly developed at lab scale, while our study, to the best of our knowledge, reports the first evaluation of the industrial-scale application of the emerging technology, providing important insights to all pyrometallurgical refinement technologies. The hydrometallurgical processes in Redwood that refine both energized batteries and scrap are analogous to the prevailing hydrometallurgy, and our analysis identifying environmental impact contributors to the Hy process similarly advances Hy processes beyond Redwood. The manuscript

has been revised to better generalize the working conditions of Redwood technologies (lines 262–267 and 513–528), and compare with existing refinement methods (lines 531–534):

“Similar to RC, emerging oxygen-free pyrometallurgical processes rely on carbothermic and thermite reduction for recycling cathode metal compounds at moderate temperatures (600–1000°C)¹⁹⁻²¹. Our results show that the RC step accounts for 5.5%–7.4% of the total environmental impacts in Redwood (Fig. 3), demonstrating the environmental feasibility of industrial-scale carbothermal reduction-based pyrometallurgy.”

“Disaggregated analysis of LIB refinement pathways at Redwood Materials ~~provides~~ provided important insights into the performance and potential of different refinement processes. Pyrometallurgical technologies are advantageous in scalability and operating safety^{9,10,33}, ~~While pyrometallurgical processing is but are~~ widely considered as ~~more~~ environmentally intensive ~~than hydrometallurgy~~^{31,32} due to the high reaction temperature. Oxygen-free carbothermic reduction and thermite reduction have been recently investigated in lab scale for energy-efficient LIB cathode pyrolysis¹⁹⁻²¹. Our analysis showed that ~~Redwood Materials’ industrial-scale carbothermal reduction-based RC-pathway pyrometallurgy (reductive calcination in Redwood Materials)~~ exhibits much lower environmental impacts than ~~prevailing direct roasting and smelting pyrometallurgical and current Hy-containing hydrometallurgical pathways reported in practice and in literature (Fig. S3). The optimized conditions of RC processing minimizes the combustion of carbon-containing LIB materials, significantly reducing CO₂-eq emissions while simultaneously generating products that are amenable for hydrometallurgical separation.~~ Our findings showed the promise of carbothermal reduction-based pyrometallurgy for pretreatment of LIB recycling; however, we note that controlling the process conditions to achieve optimal environmental impacts and product formats will elevate the costs.”

“~~Because Our analysis identified~~ chemical consumables such as H₂O₂ ~~are as~~ important contributors to hydrometallurgy, ~~indicating that~~ environmental impacts of Hy processes ~~could can~~ be reduced through more sustainable ~~(e.g., electrochemical)~~ production methods ~~(e.g., electrochemical generation of H₂O₂)³⁶.”~~

Second, operational data from Redwood Materials provide more granular insights to verify and correct the existing model-based studies. This Redwood-based study helped identify the gaps between models and industrial operations, and we highlighted the needs for advancing the whole field in the revised manuscript (line 107–114 and 536–541):

“~~Previous efforts have worked towards calculating~~ ~~There is a critical need for transparency and detailed insights into the~~ environmental impacts (e.g., energy consumption, greenhouse gas emission, and water consumption) of LIB refinement pathways and all cradle-to-gate supply chain steps. ~~Previous efforts have worked towards addressing this need^{11,12}, and this study builds on the comparative methodology of a recent step-by-step study to provide higher resolution and more actionable primary data, insights, and recommendations.~~ however, gate-to-gate analyses of circular refinement processes reported environmental impacts differing by >30%^{10,11,13} due to inconsistent methodologies.”

“We note the inconsistencies among existing studies on assessing conventional LIB refinement due to different product compositions, which underscores the need to standardize the functional unit in reporting environmental analyses for future studies in the field. The deviations between our results of circular refinement and model-based literature data highlight the importance of reconciling models with industrial operating data.”

Third, the versatile technologies at Redwood Materials enable treatment of different feedstocks (energized batteries and battery scrap) and generation of different product formats (mixed metal sulfate and discrete salts), offering opportunities to investigate influences of multiple factors on the circular refinement. Using Redwood data, we analyzed the refinement of energized batteries and scrap, providing suggestions for developing generalizable strategies for different recycling sources (**Fig. 3** and lines 542–547). Furthermore, while prevailing recycling produces discrete salts, we analyzed different refinement scenarios and demonstrated the advantages of producing mixed salts (**Fig. 4** and lines 315–320).

“Our findings provided important guidance for sources and products in future refinement. Battery scrap is currently the primary recycling source due to the proliferation of gigafactories, but will decrease in the future with quality control advances in manufacturing¹¹. While refining recycled batteries requires higher greenhouse gas emissions, energy input, and water demand than refining battery scrap (Fig. 3), it is critical to improve the technologies for recycling energized batteries for future challenges.”

*“The RC pathway (RC+Me+Hy) ~~exhibits~~ exhibited lower energy consumption (–72.3%), CO₂-eq emissions (–39.5%), and water consumption (–12%) relative to the Me+Hy pathway (**Fig. 4**), because it avoids additional treatment separating (Ni,Co)SO₄ to discrete salts. Overall, our results ~~indicate~~ indicated that refining batteries to mixed metal salts instead of discrete salts can substantially save environmental impacts while still satisfying the needs of circular LIB supply chains.”*

We have revised the introduction to better sharpen the research gaps answered by our Redwood-based analysis and importance of the findings based on Redwood data (lines 97–122 and 145–152):

~~“While several previous studies have investigated cradle-to-gate environmental impacts, gate-to-gate analyses of circular refinement processes are inconsistent, reporting environmental impacts that differ by >30%⁸⁻¹⁰, and are not completely based on industrial-scale LIB recycling operations. The gate-to-gate refinement processes utilized at established and emerging circular refinement facilities may include mechanical separation (Me), pyrometallurgy (Py)^{8,9}, and hydrometallurgy (Hy)^{10,11}. Specifically, Me physically dismantles LIBs into constituent components, Py leverages elevated temperature to facilitate thermally-driven material transformations, and Hy separates materials in the aqueous phase via leaching, precipitation, and solvent extraction processes. Variations in environmental impacts arise from the specific operational choices at refinement facilities that utilize different processing pathways and from the methods to evaluate them. Previous efforts have worked towards calculating There is a critical need for transparency and detailed insights into the environmental impacts (e.g., energy consumption, greenhouse gas emission, and water consumption) of LIB refinement pathways and all cradle-to-gate supply chain steps. Previous efforts have worked towards addressing this need^{11,12}, and this study builds on the comparative methodology of a recent step-by-step study to provide higher resolution and more actionable primary data, insights, and recommendations: however, gate-to-gate analyses of circular refinement processes reported environmental impacts differing by >30%^{10,11,13} due to inconsistent methodologies. In addition, Advancing decision-making capabilities to scale sustainable LIB supply chains requires life cycle assessment with more granular data at each step, inclusion of and incorporating industrial-scale refinement operations with practical mixed-stream battery feedstocks can uniquely inform rational design of refinement technologies. The development of LIB manufacturing and needs for the recycling market have been projected by academic and industrial researchers^{8,14}, but industrial-level understanding of the environmental influences of different feedstocks and refinement products is still lacking. documentation of operational~~

~~parameters, and qualification of results in terms of limitations and applicability to real world scenarios.”~~

“Findings of this study provide important guidance to advancing comprehensive refinement technologies. With the methodologies and results reported in this study, researchers can prioritize major opportunities to improve process efficiencies, practitioners can benchmark their environmental impacts, and policymakers can incentivize best environmental practices in LIB supply chain management. Granular insights provided by this study based on practical data can also help recyclers optimize the environmental impacts of their refinement processes, and spark more academic-industrial collaborations to further advance the field.”

Reviewer: *For (2), Since several studies have already looked at both primary and secondary battery supply chains and compared the impacts of the various process steps (even if they did not categorize it as extraction and refining, those steps are implicit in the reference process description) and across different environmental metrics, I would like more clarity on how and why this work is different, and why that difference is important.*

Our response: Regarding the stepwise cradle-to-gate analysis, we found in our literature review that different reports combine parts of the battery production chain in unique fashions, which leads to opaque comparisons (both comparing different steps and non-reproducible analyses). While these steps may be implicit, they are not uniformly categorized nor quantified across reports, which challenges comparison and reproducibility. Our contribution to the literature is a clear, consistent delineation between extraction, transport, and refinement steps in both the conventional and circular supply chains. In particular, our addition of granular industrial recycling data provides additional insights that cannot be captured by simply modeling processes, particularly for mixed-stream feedstocks and mixed-salt products. More broadly, critical insights arise from attributing and comparing the relative environmental impacts to each of these steps across circular and conventional supply chains. We anticipate that our rigorous, systematic approach can help standardize future literature especially as battery recycling scales rapidly in the near future.

Reviewer: *Lines 153 – 154, 286-287, ...: Some of the major study findings, such as that “recycling exhibit lower environmental impacts than primary production” or that “electricity consumption dominates recycling environmental impacts” have already been established in previous studies. I think the authors need to make a clearer case of what their results add to our current understanding. For instance, are they providing an important update to the magnitude of the comparative differences, or greater clarity about relative impacts, or a more granular level of detail not available in previous studies but which are potentially important? For example, something I don’t remember in other studies is an analysis like that shown in Figure 5b, which seems like an outcome that could be elevated and used as a basis for the broader discussion that incorporates consideration for (grid) energy transitions.*

Our response: We thank the reviewer for raising the issue of novelty. Below we first explain the important new insights provided by the electricity analysis, along with further revisions made to the manuscript to highlight the novelty.

We acknowledge that it can be intuitive that electricity affects the environmental impacts of LIB supply chains, but point out that our study is the first to assess comprehensive industrial recycling

pathways developed outside of East Asia, and provides important findings in addition to the electricity-carbon intensity relationship.

First, the majority of previous studies that investigated the influence of electricity on battery recycling reference specific patent applications (e.g., Umicore¹ and Brunp² for hydrometallurgy), model data gaps with assumptions (i.e., using data from similar industrial processes outside of battery manufacturing³), or receive incomplete industry data through surveys⁴. In particular, we note that many existing analyses refer to a 2015 study by Xie et al.², where the input/output of recycled batteries was adapted from a Chinese company but recycling technologies were not explicitly reported nor explicitly based on industrial-scale operations. Therefore, for the first time the current work employs comprehensive industrial-scale lithium-ion battery recycling data and processes developed in the U.S. to **quantitatively identify the roles of input electricity at a balancing area specific for comprehensive environmental impact metrics** (i.e., CO₂-eq, water consumption, and criteria air pollutants). Our results provide benchmark electricity contributions to environmental impacts for state-of-the-art lithium-ion battery recycling technologies.

Second, our in-depth analysis of environmental impacts of energy sources indicates a tradeoff relationship between CO₂-eq emissions and water consumption, which are important contributions to the broader audience of *Nature Communications*. Importantly, water consumption's connection in the broader context of LIB supply chains has not been interrogated nor summarized for a broad audience.

We are also grateful to the reviewer for highlighting our important contributions. We added lines 125–133, 133–135, 135–137, 138–140, and 140–152 to the main manuscript to better clarify the novelty of the findings:

“First, we ~~quantify~~ quantified and compare...Industrial-scale operational data provided by Redwood Materials are analyzed from a more granular level and compared to conventional LIB supply chain values based on Argonne National Laboratory’s Greenhouse Gases, Regulated Emissions, and Energy use in Technologies (GREET 2021) model¹².”

“Second, for the first time we quantitatively identified the roles of input electricity at the balancing area level on comprehensive environmental impact metrics based on industrial-scale LIB recycling data.”

“~~Second~~Third, influences of the product formats in the refinement pathways on environmental impacts ~~are~~ were examined by varying the industrial-scale refinement pathways.”

“~~Third~~Fourth, we ~~assess~~ assessed the environmental impacts of upstream processes before gate-to-gate refinement based on modeling.”

“With the methodologies and results reported herein, researchers can prioritize major opportunities to improve process efficiencies, practitioners can benchmark their environmental impacts, and policymakers can incentivize best environmental practices in LIB supply chain management. Granular ~~fi~~nsights provided by this study based on practical data can ~~also~~ help recyclers optimize the environmental impacts of their refinement processes, and spark more academic-industrial collaborations to further advance the field.”

References

- 1 Cheret, D. & Santen, S. (Patent No. US007169206B2). <https://patents.google.com/patent/US7169206B2/en>, 2007).
- 2 谢英豪, 余海军, 欧彦楠 & 李长东. 废旧动力电池回收的环境影响评价研究. *无机盐工业* **47**, 43-46 (2015).
- 3 Brown, H. L. *Energy analysis of 108 industrial processes*. (The Fairmont Press, Inc., 1996).
- 4 Boyden, A., Soo, V. K. & Doolan, M. The environmental impacts of recycling portable lithium-ion batteries. *Procedia Cirp* **48**, 188-193 (2016).
- 5 Yu, L., Bai, Y., Polzin, B. & Belharouak, I. Unlocking the value of recycling scrap from Li-ion battery manufacturing: Challenges and outlook. *J. Power Sources* **593**, 233955 (2024).
- 6 Gaines, L. & Wang, Y. How to maximize the value recovered from Li-ion batteries: hydrometallurgical or direct recycling? *Electrochem. Soc. Interface* **30**, 51 (2021).

Reviewer: 201 – 204, Figure 3, Figure 5,...: *While it is interesting that the authors include Recycled scrap in the analysis, I would expect some additional context for how to interpret its relevance. For instance, what are the relative material flows of scrap vs recycled batteries in secondary supply chains (currently or projected for the future)? For example, if scrap-to-recycled battery stream ratio is 1:100, then the additional reduction in water consumption from 74% for battery streams to 88.4% for scrap does not really amount to much, and is not particularly important to designing strategies for water efficiency and resilience in circular supply chains.*

Our response: We thank the reviewer for raising the valuable point of the relevance of battery scrap. Battery scrap is generated during battery manufacturing and assembly as byproducts, wastes, and defective batteries. While advances in manufacturing are expected to reduce the scrap rate to <10%, the rapid growth of new production lines by rising gigafactories will produce a large amount of scraps in the next decade^{5,6}. Battery scrap is now considered as a primary and ideal proximate source for recycling, and is projected to account for approximately half of the total recycling source by 2030^{5,6}. Therefore, considering both recycled scrap and batteries as major sources is necessary (particularly in the near term), and is an important contribution to the field from our paper.

The manuscript has been revised to better clarify the considerations and significance of battery scrap (lines 159–163 and 543–549):

“Battery scrap generated from manufacturing and assembly is considered a primary recycling source today, and is projected to account for approximately half of the recycling source material in the next decade as battery production outpaces disposal of energized batteries^{8,14}. Therefore, circular refinement was analyzed ~~starts~~ starting with either end-of-life batteries (1 and 2) or battery scrap (5).”

“Battery scrap is currently the primary recycling source with the rising gigafactories, but will decrease in the future with quality control advances in manufacturing¹¹. While refining recycled batteries is more environmentally intensive than refining scrap (Fig. 3), it is critical to improve the technologies for recycling energized batteries for future challenges. Our findings also demonstrated the benefits of mixed metal sulfates as refinement products over single salts in the circular supply chain, indicating that the further separations between Ni and Co salts can be avoided.”

Reviewer: *Additional Comments:*

Figure 1: What is the distinction between a conventional “material refinery facility” and a circular “recycling facility” in Figure 1b?

Our response: We thank the reviewer for pointing out the ambiguity. In general, the conventional “material refinery facility” and circular “recycling facility” are both refinery facilities. However, compared to a conventional material refinery facility, a circular recycling facility differs in the 1) feedstocks received and 2) refining technologies employed. The ore concentrate from mining enters material refinery facilities, whereas energized batteries or battery scraps are the feedstocks for a circular recycling facility. Conventional facilities employ multistep refining methods, and a circular facility utilizes technologies described in this study, including pyrometallurgy, mechanical processing, hydrometallurgy, etc.

To avoid potential confusion, we renamed the “material refinery” and “recycling facility” to “mining refinery” and “recycling refinery”, respectively, and have revised Fig. 1b to:

We also added lines 83–90 to the caption of Fig. 1b to further clarify:

“Cradle-to-gate steps of manufacturing battery-grade LIB materials (i.e., salts) from conventional and circular supply chains, both of which include three steps: extraction, transport, and refinement. Conventional extraction refers to natural mining, and the circular counterpart is battery collection. Transport in the conventional and circular supply chains move ore concentrate and batteries, respectively. Conventional mining refineries and circular recycling refineries receive ore concentrates and batteries, respectively, and employ different refining technologies. Extraction and transport are considered “upstream” steps relative to gate-to-gate refinement...

Reviewer: Figure 2 – Process descriptions: It would help if the authors included (perhaps as supplemental material) the details of the underlying process descriptions for the different pathways illustrated, specifically to help with distinguishing between Redwood process and “Representative Circular (RC)” pathway. For instance, do they share common/similar process

steps and are the assumptions about these overlapping steps identical? Do they use the same chemicals? What degree of chemicals/solvent recovery or reuse is assumed? Does the RC Hy process (in Figure 3) also include mechanical processing - which one would expect, but which is not obvious from the figure?*

Our response: The reductive calcination (RC) process at Redwood falls under the comprehensive category of “pyrometallurgical technologies”. RC at Redwood is the industrial-scale application of the emerging carbothermal reduction-based pyrometallurgy, and employs optimized operating conditions. We have revised the manuscript to better distinguish the technology (lines 247–255), but cannot share operating details like chemical recovery or reuse due to non-disclosure agreements with Redwood. We aim to achieve a compromise in this academic-industrial collaboration between academic transparency and industrial intellectual property. We consider this compromise critical to provide access to data that facilitate the granular insights we report in this study, and as an opportunity to model academic-industrial collaborations in the rapidly developing battery recycling field.

“~~The RC process is an innovative exothermic pyrometallurgical process that converts~~ reduces the cathode metal oxide compounds under the oxygen-free condition energized battery feedstock under certain conditions that leverage heat from exothermic processes and inhibit graphite combustion for subsequent refinement. Unlike dominant industrial pyrometallurgical processes (e.g., direct roasting or smelting) that require high temperature >1400 °C, ~~This the RC process does not use~~ optimizes the working conditions, favoring the carbothermal reduction without using graphite carbon, thus avoiding direct fossil fuel inputs ~~onsite and facilitates subsequent hydrometallurgical refinement into battery-grade materials and graphite combustion.~~”

We confirm that the representative hydrometallurgy (Hy* in Fig. 3) includes a mechanical processing step. Thus, an analogous Me+Hy process is modeled to analyze the environmental impacts of producing discrete salts in Fig. 4. We have revised the manuscript to describe this point more explicitly (lines 257–260 and 310–313):

“Energy consumption and CO₂-eq emissions of representative existing recycling pathways from the literature, including pyrometallurgy (Py), hydrometallurgy (Hy*), and direct recycling (Direct*), ~~are~~ were also presented in Fig. 3 for comparison. Note that a mechanical processing step is often included in existing Hy* recycling.”*

“Circular pathways refining batteries to different products ~~are~~ were analyzed using the Redwood data by the RC+Me+Hy process and the modeling of a representative battery recycling method combining mechanical and hydrometallurgy (Me+Hy, analogous to the Hy in Fig. 3) refinement (Method (3) in Fig. 2).”*

Reviewer: *Figure 3 caption: Please correct the **b** and **c** captions of Figure 3 – it appears they are switched.*

Our response: We thank the reviewer for pointing out the error and have revised Fig. 3 with data correction:

Reviewer: Lines 234 – 249: *It is good to see that the authors highlighted these differences in predicted LCA impacts between studies, even if just for reference. However, how exactly should we interpret the results from the current work against the backdrop of those other results? Are there obvious limitations in other studies that is corrected for here? Can we have greater transparency in the underlying process and data for the current work?*

Our response: We thank the reviewer for favorable comments on our findings and raising the issue of transparency in the field of battery recycling.

First, we have substantially validated our analytical methodology by comparing the findings to the existing literature data of both conventional and circular supply chains. According to the comparison, our results are reliable (lines 197–199 and 260–261):

“The values of ~~energy consumption and~~ greenhouse gas emissions ~~are~~ were comparable with previous studies based on GREET datasets^{12,15} (Fig. S3b).”

“In general, the RC+Me+Hy pathway at Redwood exhibits exhibited comparable energy consumption and CO₂-eq emissions with Hy and Direct literature values¹¹”

Second, we attributed the differences in LCA results to *divergent life cycle assessment approaches, processing conditions, and the utility of primary industrial data access over modeling processes from literature sources*, and have mentioned the corresponding limitations in the main manuscript (lines 278–282, 282–285, and 285–288). We further note that the differences highlight the novelty and importance of the current work, i.e., collating real operating data with the model-based results.

“However, direct comparison can be inexact due to varying underlying assumptions and data sources. For example, Argonne National Laboratory’s GREET and EverBatt models leverage a combination of technology descriptions from patent applications, literature data on process flow consumptions, industry site visits and surveys, expert advice solicitation, and stated assumptions to form complete pathways.”

*“Further, Ciez and Whitacre quantified environmental impacts using output products represented as “metal offsets” for pyrometallurgy or with metals in solution for hydrometallurgy¹¹ (Note 3 in **Supplementary Information**), rather than cathode salts in this study.”*

“In addition, the previous studies included a portion of recycled metal materials in its conventional supply chain analysis, whereas this work references only mined natural deposits in conventional supply chains to fully deconvolute the environmental impacts¹².”

Third, to overcome the limitations of previous studies and maximize the transparency of the current work, we have clearly described our method and provided a full set of data in Supplementary Information and Supplementary Data File. We note that the refinement technologies by Redwood Materials in this paper is innovative, and some working details are not explicitly published, to balance between novelty and transparency.

Lastly, we acknowledge that *standardization* of methodology and communications is critically needed to advance the whole research field of battery recycling. We have added lines XX for further discussion, and highlighted the idea in the introduction (lines 536–541, 145–146, and 150–152):

“We note the inconsistencies among existing studies on assessing conventional LIB refinement due to different product compositions, which underscores the need to standardize the functional unit in reporting environmental analyses for future studies in the field. The deviations between our results of circular refinement and model-based literature data highlight the importance of reconciling models with industrial operating data.”

“Findings of this study provide important guidance to advancing comprehensive refinement technologies.”

“Granular insights provided by this study based on practical data can also help recyclers optimize the environmental impacts of their refinement processes, and spark more academic-industrial collaborations to further advance the field.”

Reviewer: Lines 258 – 260: *The authors suggest that in conventional supply chains, producing mixed salts has higher energy and carbon footprint compared to producing discrete salts. This result is counter-intuitive, since I'd expect that producing individual salts would theoretically*

include an additional separation step, which adds energy, and consequently, CO₂ emissions. Could the authors explain this non-intuitive trend, and how come it is different from that seen for the circular pathway? It would help if they also provide the underlying process description as well.

Our response: We are grateful to the reviewer for raising the confusion. According to the GREET model, producing discrete salts is not achieved from post-treatment of mixed salts, but starts from ores distinct from the laterite refined for mixed salts. In conventional refinement, the discrete salts, NiSO₄ and CoSO₄, are refined from Ni-rich and Co-rich ores separately, generating little mixed salts except impurities. On the other hand, producing the mixed hydroxide precipitate (Ni,Co)(OH)₂ starts from refining the Ni laterite, where the concentration of Co is very low (3.6%), thus significantly elevating the energy consumption.

We have revised the manuscript to provide a clearer explanation (lines 302–309):

“While the discrete products NiSO₄ and CoSO₄ are produced from the ~~mixed hydroxide precipitates through additional post-treatment~~ Ni-rich and Co-rich ores separately, generating mixed hydroxide precipitates starts from the Ni laterite where, the very low composition of Co (3.6%) ~~in the latter~~ limits the NCA stoichiometry, thus increasing the total energy cost to generate 1 kg NCA-equivalent materials. On the other hand, water consumption of refining mixed hydroxides is slightly lower (–6.6%) than that in producing discrete salts. In general, discrete salts-based pathway is was favorable in conventional refinement to reduce environmental impacts.”

Reviewer: *Figure 6: I am not sure why Figure 6 is presented as a result, and not as supplemental information illustrating the approach used to model the transport logistics.*

Our response: The contents have been combined with the original Fig. 7 in the revised manuscript (see next page). Data in the bar graphs of original Fig. 7 have also been updated based on earlier corrections that were not carried through from earlier figures.

We have also revised the caption correspondingly (lines X).

“Fig. 6 | ~~A logistics model for assessing upstream environmental impacts of extraction and transport in circular and conventional lithium-ion battery supply chains.~~ Cradle-to-gate environmental impacts of different supply chains. A, Modeled circular extraction of LCO-based smartphones from every census-block group based on population to the closest, existing private or municipal collection facility (CF) using a shortest-route algorithm. Inset details modeled circular transport of smartphones aggregated at CFs and then transported to a central recycling facility at the center (gravity point) of the California population by the shortest route (red lines). Colors of block groups indicate the catchment area of a specific CF, where CF size shows the relative number of smartphones collected in 2019. B, A weighted distribution estimate of international transport logistics for conventional supply chains between mining and refining countries based on cobalt productivity in the top Sankey diagram. C, An example of transport logistics for cobalt mined and aggregated in the Democratic Republic of the Congo (DRC) and then shipped via primary road,

train rail, and maritime routes using a shortest-distance path to major refinery locations, with insets showing the degree of detail considered. Similar analyses were performed for Li, Ni, Co, and Al. Inserts present more detailed transit routes in DRC and Canada. **Ad**, Energy consumption (left), **be**, CO₂-eq emissions (middle), and **ef**, water consumption (right) of conventional (conv.) and circular (cir.) supply chains by step including material extraction, transport, and refinement. NCA-eq cathode used in electric vehicles (EV-NCA, left panels) and LCO-eq cathode material used in smartphones (Phone-LCO, right panels) are provided. Environmental impacts of refinement are analyzed based on electricity generated from balancing grid authority CISO and upstream supply chain steps (extraction and transport) are based on data from GREET and transport models developed in the preceding section and depicted in **Fig. 6** panels a–c. Specific environmental impacts of each step are detailed in Tables S5–S7.”

Reviewer: *Lines 354 – 355, Figure 6: I think there is a very important element that is missing, which is scale.*

To compare primary and secondary logistics fairly, it would be good to try to make the scale more comparable. For example, expanding the collection radius from just California to say, the entire U.S. significantly changes the associated impact footprints. While I expect this to still be smaller than conventional, the comparison at this scale will be more justifiable.

The converse argument is that for a much smaller scale of primary manufacturing, it is possible to identify a domestic supply chain that ends up with a much smaller footprint than the transnational supply chain considered in the model in Figure 6.

Our response: We thank the reviewer for the suggestion. We agree that scale may influence our results, but given that battery recycling is not as widespread as battery manufacturing, there is an inherent mismatch in current scale. Our study of California represents a *localized* domestic processing strategy with lower environmental impacts instead of inter-state transport. Collecting and transporting LIBs to recycling centers inside California minimizes the distances and the environmental impacts, and can benefit safer handling LIBs. We also note that this study is focused on refinement, and analyzing the upstream steps at the scale of one state is a first step. Considering LIB recycling in the regions of multiple states can be beneficial, but warrants further investigation outside the scope of this study. We have added lines 582–587 to highlight the point:

“Upstream process optimization of environmental impacts warrants further investigation, such as the active area of high-throughput automation of LIB extraction from non-standardized devices and EV battery packs or rapid assessment of LIBs for second life uses. While the current study modeled localized collection and transport inside the state of California, extending the scale to regions combining multiple states is critical to further investigate the upstream environmental impacts.”

Reviewer: *Figure 7: Can the authors clarify the reason for the difference in transport energy and CO₂ footprint between EV and smartphone batteries? Does the difference arise from relative battery compositions, or transport distances? In addition, are there any key differences in their respective recycling steps, and how do these translate to differences in overall lifecycle impacts?*

Our response: Due to different packing densities, the mass of active materials per kg of battery for LCO (0.162) is much lower than that of NCA (37.4). Therefore, the environmental impacts of

transport LCO normalized by kg active materials are correspondingly higher. Information of the packing densities was detailed in Table S10 in the Supplementary Information, and we have revised the main manuscript to better clarify (lines 479–482):

“Circular production of LCO-grade materials leads to higher environmental impacts than that of NCA-grade materials based on the mixed-stream feedstock composition processed by Redwood Materials. Note that LCO has relatively lower packing densities of active materials compared with NCA (detailed in Table S10, thus elevating the environmental impacts of transporting LCO).”

Reviewer: Lines 485 – 487 – *Can the authors say more/ be a bit more specific here? Anyone can mention this trade-off without analysis. I would expect that the current analysis can supply enough information to recommend which of the two options – disassembly before transport vs transport before disassembly - is potentially a more impactful approach?*

Our response: We appreciate the suggestion from the reviewer, but note that concluding remarks of the environmental impacts of the two disassembling strategies are beyond the scope of the current study and warrant additional investigation. In addition, transport plays a minor role in the overall environmental impacts of circular battery supply chains (**Fig. d–f**), and the disassembling strategies will not have significantly different contributions to the overall environmental impacts. Currently there is no standardization for LIB design (packs, modules, and cells) or integration into electric vehicles (e.g., cell-to-chassis, module-to-chassis, cell-to-pack), challenging the evaluation of environmental impacts of the disassembling. Distance-to-facility and economies-of-scale for different operations would need to be considered. In addition, pre-processing materials into black mass prior to shipping to a recycling facility is also a consideration to reduce both shipping costs and lower safety concerns present for energized LIBs.

We have changed lines 572–578 to better clarify:

“Practical battery collection operations will likely vary based on route selection and preprocessing strategy further influencing environmental impacts⁶. ~~For example, the Further investigating the environmental impacts of the disassembly of collected EV battery packs or removal of smartphone batteries from devices prior to transport can help balance between to a recycling facility can increase extraction energy usage through extraction but reduce and transport environmental impacts by lowering transportation weight (Table S10).~~”

Reviewer: Lines 489 – 490: *A number of studies mention the potential of rail transport to reduce energy and environmental impacts of secondary battery supply chain logistics relative to trucking, but how practical is this, especially for this analysis which is restricted to California? I imagine there are major limitations that make this nearly impossible, such as inadequate local rail coverage, competition with other high value cargo, etc.*

Our response: We acknowledge the inadequacy of local rail coverage, but note that a rail line can be used as an aggregation point for long transport combined with other vehicles. Determination of the optimal transport strategy for a certain collection/transport task warrants investigations of specific available vehicles. Implementing policy recommendations of collection points to train depots will also facilitate the utilization of available rail transport. The manuscript has been revised to reflect this consideration (lines 578–582):

“Trucks are used as the primary vehicle for transport analysis given regulatory concerns that consider LIBs hazardous material in many transportation scenarios³⁹. However, alternative transport like railway can further lower environmental impacts by approximately four times versus trucking (Tables S12), and can be explored for use as aggregation points for long transport combined with trucking.”

Reviewer: Impact allocation method: *Could the authors clarify what type of impact allocation method they used. For instance, was it based on the embodied economic value of the target products in the recycled or primary pathways?*

Our response: Among commonly tracked sustainability metrics, including environmental metrics, financial metrics, social metrics, and governance metrics, this study is focused on understanding the environmental impact metrics of LIB recycling. We follow the environmental metrics criteria proposed by United Nations’ Sustainable Development Goals and widely accepted by the community of life cycle assessment studies, i.e., energy consumption, gas emissions, and water consumption. Target products in the conventional and circular supply chains were based on the recycling operations in Redwood Materials and the GREET model, respectively.

We added lines 163–166 and 180–183 to better clarify our impact allocation method:

“Ni and Co in refinement products for subsequent manufacturing can be discrete salts ((1) and (3)) or mixed compounds (2, 4, and 5). Target products of the conventional and circular pathways were based on the GREET model and practical recycling operations, respectively.”

*“~~Environmental impacts including e~~Energy consumption, greenhouse gas emissions (CO₂-equivalents, CO₂-eq; additional criteria air pollutants are detailed in **Table S1**) and water consumption were chosen as the key metrics to analyze environmental impacts of LIB supply chains in this study^{11,16,17}.”*

REVIEWER COMMENTS

Reviewer #2 (Remarks to the Author):

The authors have taken a major step to revise the manuscript and some of the previous concerns have been (partially addressed), however, this reviewer remains conservative of this work at this current format. A few key questions are summarized for the authors' consideration:

1. It is of great concern to keep emphasizing Redwood RC technology and respectively exaggerate one specific company. It is known that RC is not exclusive for Redwood and they are not even the first company doing this. This reviewer understood that some of the data cited in this study are from Redwood, but the more data used for comparative analysis are cited from other literature, sometimes 2019, sometimes 2021, and 2023 crossing different segments, which causes inconsistency and confusion. We must realize that keeping a neutral opinion and be fair is the fundamental importance for academic integrity. Through the whole manuscript, this reviewer has a feeling that the authors are mainly praising a specific tech company which is very much concerned, especially for such a journal.
2. The first part of the manuscript is more recycling analysis/comparison, and the other half of this work is on other less coherent topics such as electricity. It is understandable that the entire supply chain could be compared, however, each segment needs to be tied with the specific process. Especially, the electricity part is something quite loosely attached to the main topic and the relatively random geographic effect on the cost and environmental impact on recycling may not be directly translated into any useful judgement as the recycling innovation has little to do with the power company (their electricity generation process).
3. It should be noted that mixed hydroxide product provides more value and embedded energy (as well as CO₂ emission) compared with mixed metal sulfate. To close the recycling loop, sulfates will need to be processed further with consuming energy and CO₂ emission to produce hydroxide before cathode materials can be made. So the overall analysis of environmental impacts of "conventional" and circular refining technologies should be more fairly compared, otherwise the conclusion is misleading.
4. The authors claimed that shipping is a minor factor for cost and CO₂, but transport of spent batteries is a different story. The authors still need to address this point.
5. The current manuscript, after revision, is an assembly of many pieces and some of them are a little out of scope for recycling and supply chain of batteries. It gives an impression like a working report instead of a tight/coherent analysis around a central topic. Some of the less relevant context might not be necessary and can be removed to make this manuscript tight and lean.

Reviewer #3 (Remarks to the Author):

Please see attached document for comments

Review Summary

This paper presents a comparative life cycle analysis for primary and secondary supply chains for lithium-ion batteries. As stated in my previous review, the paper remains well written, the topic is of interest, the core methodology is sound, and outcomes relevant for advancing the conversation around designing circular supply chains for lithium-ion batteries. After reviewing the author responses to my comments and updates to the manuscript, I think that:

- 1) The authors have adequately addressed the issues and questions I raised and have consequently updated the manuscript.
- 2) The updates to the manuscript have added clarity and context for the presented results, and clarified the nature and uniqueness of the main contributions from this work.
- 3) While I understand the sensitivities around industrial data, it remains a pity that this constrains transparency and limits the ability for benchmarking across different analyses, as well as drawing certain broader conclusions from reported outcomes. That said, I agree that this represents a major step in the right direction.
- 4) I believe this study makes an important contribution to foundational conversations around circular supply chains for battery. There is certainly more work required in terms of informing policy and strategy for incentivizing circular battery supply chains. It will be good to see analyses that incorporate adequate projected scale of circular supply chains, as well as economics of battery recycling, and quantitatively assess the potential impact (environmental, economic, social) of any number of policy interventions (e.g., producer responsibility, collection programs) and strategies (e.g. design for efficient disassembly, disaggregated siting of facilities for different recycling steps).
- 5) I recommend this submission for publication.

Minor comment

Lines 524 – 527: “... *optimal environmental impacts and product formats will elevate costs.*”

- 1) Is this demonstrably the case, or simply intuition based on likely design/operational interventions? I imagine certain adjustments to process conditions imply cost/environmental impact trade-offs, but it is not necessarily general.

Responses to Reviewers

Manuscript Title: Life cycle comparison of industrial-scale lithium-ion battery recycling and mining supply chains

Manuscript ID: NCOMMS-23-46463A

In this document, we respond point-by-point to comments from the reviewers and the editor on our original manuscript submission. We thank the reviewers for their insightful feedback, and have incorporated several of their suggestions to further improve the manuscript before publication. The legend for this document is as follows:

Reviewer comments are in italics.

Author responses are in standard type and indented.

Deletion and addition to the original manuscript are marked in red and blue texts, respectively.

Line numbers are specified for either the original manuscript or the revised manuscript.

■ REVIEWER 2

Reviewer: *The authors have taken a major step to revise the manuscript and some of the previous concerns have been (partially addressed), however, this reviewer remains conservative of this work at this current format. A few key questions are summarized for the authors' consideration:*

Our response: We are grateful to the reviewer for the favorable comments on our revision efforts, and have provided further improvements to address the questions raised. Specific responses are provided below and major changes are:

- i. updating the data for environmental impacts of electricity to enhance the reference consistency, and inclusion of a summary of types of data use;
- ii. clarifying that the RC technology is not exclusive to Redwood, but has comprehensive implications for carbothermal reduction-based pyrometallurgy at industrial scale;
- iii. highlighting the focus and structure of result presentation, sharpening the discussion, and enhancing the readability of the manuscript.

Reviewer: *1. It is of great concern to keep emphasizing Redwood RC technology and respectively exaggerate one specific company. It is known that RC is not exclusive for Redwood and they are not even the first company doing this. This reviewer understood that some of the data cited in this study are from Redwood, but the more data used for comparative analysis are cited from other literature, sometimes 2019, sometimes 2021, and 2023 crossing different segments, which causes inconsistency and confusion. We must realize that keeping a neutral opinion and be fair is the fundamental importance for academic integrity. Through the whole manuscript, this reviewer has a feeling that the authors are mainly praising a specific tech company which is very much concerned, especially for such a journal.*

Our response: We well understood the concerns of the reviewer, but politely note that we did not intend, nor feel obliged to, specially endorse the RC technology of Redwood. RC was studied as a representative carbothermal reduction-based pyrometallurgy with critical industrial-scale data. We are well aware of the development and applications of carbothermal reduction elsewhere, and intentionally did not claim RC is exclusive to Redwood. However, to the best of our knowledge, we did notice that 1) most of the existing studies on carbothermal reduction pyrometallurgy were based on lab scale, and 2) only very few companies reported the practical installation and operation, and often without accessible data for quantitative evaluations. We politely point out that analyzing the RC process at Redwood is of critical importance for the whole area of battery recycling. In addition, we find it most rigorous and accurate to only comment on the industrial data that we could access, which was from Redwood's RC process. Otherwise, we would risk over-generalizing our LCA results to other pyrometallurgical installations that may exhibit different environmental impacts.

Therefore, we highlighted the insights from the analysis for the environmental feasibility of *industrial-scale carbothermal reduction-based pyrometallurgy* (lines 269–271 and 279–283), rather than *the RC of Redwood*.

“Similar to RC, emerging oxygen-free pyrometallurgical processes rely on carbothermic and thermite reduction for recycling cathode metal compounds at moderate temperatures (600–1000°C)¹⁹⁻²¹.”

*“While this analysis was focused on Redwood Materials refinement pathways, the methodology can be used to evaluate additional refinement pathways (e.g., analysis of ~~the~~ a representative hydrometallurgy was detailed in **Fig. S3c**), or others that use different material feedstocks, refinement processes, and energy supplies.”*

To reinforce the comprehensive importance of our study, we have better highlighted the category of pyrometallurgy that the “reductive calcination” (RC) process belongs to. We also clarified our use of “Redwood” as the data source, instead of a specific name of the methods (e.g., RC+Me+Hy) (lines 254–258, 267–269, 271–276, 320–327, 355–358, 485–487, 511–513, 528–530):

*“To produce battery-grade cathode materials, Redwood Materials used a combination of reductive calcination (RC) pyrometallurgical, mechanical (Me), and hydrometallurgical (Hy) LIB refinement processes (pathways detailed in **Fig. S2**). RC is an ~~innovative~~ industrial-scale exothermic pyrometallurgical process that reduces the cathode metal oxide compounds under oxygen-free conditions for subsequent refinement.”*

“In general, the RC+Me+Hy pathway ~~at Redwood~~ exhibited comparable energy consumption and CO₂-eq emissions with Hy and Direct literature values¹¹, and substantially lower environmental impacts than Py.”*

*“While most of the carbothermic and thermite reduction processes were investigated at the lab scale²⁴⁻²⁶, ~~Our~~ our results showed that the RC step accounted for 5.5%–~~7.47.5~~7.5% of the total environmental impacts in Redwood the circular refinement (**Fig. 3**), demonstrating the environmental feasibility of industrial-scale carbothermal reduction-based pyrometallurgy. Note that traditional pyrometallurgy and Redwood Material's reductive calcination RC pyrometallurgy can process energized batteries of varying states of charge, health, and formats with minimal modification...”*

“Circular pathways refining batteries to different products were analyzed using the ~~Redwood data by the RC+Me+Hy process~~ data and the modeling of a representative battery recycling method combining mechanical and hydrometallurgy (Me+Hy, analogous to the Hy in Fig. 3) refinement (Method (3) in Fig. 2). The ~~Redwood process RC+Me+Hy pathway~~ refines recycled batteries to mixed metal sulfate, (Ni,Co)SO₄, whereas the representative Me+Hy produces discrete NiSO₄ and CoSO₄. ~~The RC pathway (RC+Me+Hy) Refining into mixed metal sulfate~~ exhibited lower energy consumption (-72.3%), CO₂-eq emissions (-~~39.5~~ 41.4%), and water consumption (-~~12~~ 8.0%) than the Me+Hy pathway (Fig. 4)...”*

“Unlike conventional pyrometallurgical processes that require external energy sources^{11, 23}, the RC ~~process~~ pyrometallurgy is primarily autothermic because it leverages process heat released from exothermic reactions of the LIB materials^{30,31}.”

“Circular production of LCO-grade materials led to higher environmental impacts than that of NCA-grade materials based on the mixed-stream feedstock composition ~~processed by Redwood Materials~~ analyzed in this study.”

“Disaggregated analysis of LIB refinement pathways ~~at Redwood Materials~~ using industrial data provided important insights into the performance and potential of different refinement processes.”

“The alternative direct recycling technology is reported to exhibit comparable environmental impacts to ~~Redwood Materials~~ circular refining methods in this study¹², but warrants further assessment after industrial-scale implementation.”

We appreciate the reviewer for pointing out the possible confusion due to the different reference years of data. The reference years were chosen based on the accessibility of prioritized data (i.e., the operational data at Redwood) and information from the literature. Note that the varied years of datasets further supports the utility of the submitted manuscript in standardizing and updating the battery recycling LCA literature. Overall, all data for analysis presented in the main manuscript were based on 2021. 2019 data were obtained from the literature for comparison, and 2023-based results were presented as Supplementary Data as requested by the reviewers in the previous submission. We have updated the data of electricity-embedded environmental impacts and modeling information to 2021 in the revised manuscript to enhance the consistency. Note that no substantial changes were observed to the conclusive remarks. Thus, the only datasets in the revised manuscript that are not 2021 are 1) the literature data, which we cannot control and 2) the 2023 GREET model as requested by the reviewer (referenced in supplemental information). To help readers catalogue these differences, we also added an additional **Table S16** to summarize the sources and use of data from different reference years, mentioned in lines 174–175.

“Unless specifically noted, all major analyses were based on 2021 data (data reference years were summarized in Table S16).”

“Table S16. Summary of sources from different reference years and how they were used in this study. Data uses were categorized into three types to represent analysis result in the main manuscript (type I), literature data comparison in the main manuscript (type II), and supplementary information for additional reference (type III).”

#	Supply Chain Step	Data Source for Presentation	Reference Year	Presentation of Data	Type of Use
1	Refinement	Redwood operational data for analyzing circular refinement	2021	Figs. 3–4	I

2		Environmental impacts of electricity for analyzing circular refinement	2021	Figs. 3–5	I
3		GREET model for analyzing conventional refinement	2021	Figs. 3–4	I
4		GREET model for analyzing conventional refinement	2023		III
5		Literature data of representative refinement technologies, for comparison	2019	Fig. 3	II
6	Extraction	For modeling	2021	Fig. 6	I
7	Transport	For modeling	2021	Fig. 6	I

Figs. 3–6 in the revised main manuscript were updated with 2021-based environmental impacts of electricity:

Fig. 3

Fig. 4

Fig. 5

Fig. 6

Reviewer: 2. *The first part of the manuscript is more recycling analysis/comparison, and the other half of this work is on other less coherent topics such as electricity. It is understandable that the entire supply chain could be compared, however, each segment needs to be tied with the specific process. Especially, the electricity part is something quite loosely attached to the main topic and the relatively random geographic effect on the cost and environmental impact on recycling may not be directly translated into any useful judgement as the recycling innovation has little to do with the power company (their electricity generation process).*

Our response: We politely note that the whole manuscript is explicitly centered around understanding environmental impacts of battery recycling and conventional mining, with the *refinement step* as our focus. In the Results section, analyses of different aspects of refinement were first presented, followed by modeling on the upstream steps and overall considerations of the whole supply chain. Centering around the environmental impacts of refinement, we presented 1)

analysis pathways (**Fig. 2**), 2) overall environmental impacts of the refinement step (**Fig. 3**), 3) impacts of product formats on the refinement step (**Fig. 4**), and 4) contributions of different input materials (including electricity) to the refinement environment impacts. All the four aspects are critical to understanding the environmental impacts of the refinement step, placed logically, and explicitly reflected by the subsection titles in the main manuscript (lines 185–186, 301, and 342–343):

*“**Refining** lithium-ion batteries into battery-grade materials exhibited lower environmental impacts than production from mined natural materials.”*

*“Formats of **refinement** products influenced environmental impacts.”*

*“Electricity consumption dominated the environmental impacts of lithium-ion battery circular **refinement**.”*

The electricity section served as a further identification of the major contributor to our main topic, **refinement environmental impacts**, and provided a systematic investigation of the influences of power sources (lines 348–351). We also politely note that varying the electricity sources is *not* presenting the *relatively random geographic effects*, but used representative electricity grids to form a gradient of CO₂-eq intensities (lines 389–391).

*“Electricity consumption was found to be a principal factor dominating the environmental impacts. For both LIB feedstock pathways (Methods (4) and (5) in **Fig. 2**), electricity accounted for 70.3–91.0% of the total energy consumption, ~~71.8–79.1~~ 70.5–83.4% of the total CO₂-eq emissions, and ~~54.3–63.6~~ 56.9–66.1% of water consumption (**Fig. 5a**).”*

*“Because electricity dominated the environmental impacts of LIB recycling processes, we compared several electricity grid balancing areas that emit a range of CO₂-eq emissions per MWh (averaged for ~~2019~~ 2021)^{32,34} in **Fig. 5b**...”*

Below we tabulate the electricity grid areas used in this study following an order of increasing embedded CO₂-eq, for the reviewer’s reference. While NEVP is where Redwood based on (highlighted), electricity grid areas of both lower (BPAT and CISO) and higher (WACM) embedded emissions were used in our methodology.

Electricity Grid	Embedded CO ₂ -eq (kg/MWh)	Normalized by NEVP
BPAT	92.8	0.2
CISO	275.8	0.6
NEVP	455.9	1.0
WACM	682.0	1.5

Our analyses on electricity were not only closely attached to the main topic, but also provided important actionable guidance to battery recycling companies. Although advice on power generation is beyond the scope of this study, our results showed that **locating the recycling facility and choosing electricity sources is critical to reducing and balancing environmental impacts of the circular refinement step in battery recycling** (lines 392–396, 545–549, and 585–588).

Based on our finding, Redwood Materials strategically changed from conventional Nevada grid electricity to electricity from Nevada renewable energy tariff (NV*) to optimize their CO₂-eq emission. Further, our results also inform other recycling companies about which part of their environmental impacts are based on electricity source versus other consumables (both for RC pyrometallurgy and for hydrometallurgy for energized, end-of-life batteries). By segmenting emissions sources, recycling companies can focus on both electricity source and innovate on technologies and inputs to further reduce their environmental impacts.

“Substituting NEVP electricity with other balancing areas including Bonneville Power Administration Transmission (BPAT), California Independent System Operator (CISO), Western Area Power Administration of Colorado-Missouri (WACM), and a 100% renewable energy tariff in Nevada (NV), yielded a significant reduction in CO₂-eq emissions of up to 93.3% (recycled scrap) and 87.4% (recycled battery) relative to conventional refinement (Fig. 5b).”*

“Electricity greatly influenced environmental impacts in LIB circular refinement, and the variability among grid electricity sources elucidated a tradeoff between CO₂-eq emissions and water consumption (Fig. 5). Therefore, considering water consumption and CO₂-eq emissions is necessary for selecting recycling facility locations, particularly in water-sensitive or emissions-sensitive scenarios.”

“As the prevalence of LIBs grows in the mobility sector and beyond, strategic placement of domestic LIB collection, refinement and manufacturing facilities can further minimize future environmental impacts by considering heterogenous LIB growth by location, collection approach, transportation distance, and electricity source for refinement processes.”

Overall, we believe the analyses of electricity were highly necessary and germane to the main topic. We have added lines 122, 132–134, 158–159, 171–175, 301–302, and 343–344 to the revised Introduction to better overview the coherence and organization of the results:

“In this study we quantified the cradle-to-gate environmental impacts of battery-grade cathode material salts manufactured in conventional and circular supply chains across three major steps: material extraction, transport, and refinement (Fig. 1b), focusing on the refinement step.”

“~~Second, for the first time we~~ This use of industrial-scale recycling quantitatively identified the ~~roles~~ dominant role of input grid electricity in circular refinement at the balancing area level on ~~comprehensive~~ environmental impact metrics based on industrial-scale LIB recycling data.”

“In this study, analyses of environmental impacts were presented with a focus on the refinement step, followed by analysis of the upstream steps in the complete LIB supply chains. In LIB supply chains, the refinement step converts the collected feedstocks into battery-grade salts for further manufacturing (Fig. 2a).”

“Target products of the conventional and circular pathways were based on the GREET model and practical recycling operations, respectively. In the following sections, the overall refining environmental impacts were first analyzed, followed by influences of product formats on the refinement step and key contributors to the refining environmental impacts. Lastly, upstream environmental impacts were analyzed and compared to the refinement step. Unless specifically noted, all major analyses were based on 2021 data (data reference years were summarized in Table SI6).”

“Product format is an important factor in understanding and properly comparing LIB refinement pathways (Fig. 2). Ni and Co are key elements for battery manufacturing, and can be traded in the

format of mixed metal salts or discrete salt products between battery refiners and battery manufacturers^{28,29}.”

*“To further understand the performance limiting factors in the refinement step, ~~The~~ the relative environmental impacts of input consumables (e.g., energy, water, commodity chemicals) in the gate-to-gate refinement processes were disaggregated in **Fig. 5** (additional criteria air pollutants in **Tables S2–S3, Figs. S4–S5**).”*

Reviewer: *3. It should be noted that mixed hydroxide product provides more value and embedded energy (as well as CO₂ emission) compared with mixed metal sulfate. To close the recycling loop, sulfates will need to be processed further with consuming energy and CO₂ emission to produced hydroxide before cathode materials can be made. So the overall analysis of environmental impacts of “conventional” and circular refining technologies should be more fairly compared, otherwise the conclusion is misleading.*

Our response: We politely note that our analysis on the refinement step is focused on production of battery-grade cathode materials, i.e., the precursors, for the supply chains, instead of the ready-to-use materials for direct use battery manufacturing. We have pointed this in lines 140–144 and 159–161 of the manuscript:

*“The upstream assessment includes the extraction of LIB material from conventional (i.e., mined ore) or circular (i.e., collected batteries) sources and the transport of extracted material to relevant refinement facilities for **production of battery-grade cathode materials as Li, Co, and Ni sulfate or carbonate salts**.*

*“In LIB supply chains, the refinement step converts the collected feedstocks into **battery-grade salts for further manufacturing (Fig. 2a)**.*

Battery-grade metal sulfates were produced at industrial scale and sold at Redwood Materials, and were also reported to be commonly produced and collected in the emerging market of battery recycling¹⁻⁵. We added lines 249–250 in the revised manuscript to further highlight the important roles of battery-grade metal sulfates in the battery supply chains:

*“Note that while the elemental stoichiometry was identical, the output battery-grade materials varied slightly between conventional (Li₂CO₃, NiSO₄, CoSO₄) and circular (Li₂SO₄, (Ni, Co)SO₄) supply chains (detailed in **Methods**). Metal sulfates are commonly produced and traded in the battery recycling market²⁰⁻²³.”*

In our comparison, both circular and conventional refinements were based on producing *sulfates*, and this has been summarized in **Fig. 2** in the main manuscript (pathway 1, 4, and 5). We have clearly stated our methods, and we believe that the comparison is overall fair and rigorous.

We have added lines 763–765 in the revised **Summary of study limitations** to further clarify our focus on producing battery-grade precursors instead of ready-to-use manufacturing materials:

*“The chemical formats of output products differ between the conventional and circular supply chains, but converting them to the same products would not substantially change the results due to the similarity between the cathode salts of the two supply chains (**Note 3 in Supplementary Information**). Battery-grade cathode metal sulfates were chosen as the refinement products for the major comparison, differing from the ready-to-use materials for manufacturing, but not substantially influencing the fairness of the comparison.”*

References

- 1 Van Hoof, G., Robertz, B. & Verrecht, B. Towards Sustainable Battery Recycling: A Carbon Footprint Comparison between Pyrometallurgical and Hydrometallurgical Battery Recycling Flowsheets. *Metals* **13**, 1915 (2023).
- 2 Reaugh, L. RecycLiCo Battery Materials produces 99.99% pure lithium sulfate from the RecycLiCo Process. *RecycLiCo Battery Materials* (2021).
- 3 Mcnees, M. LG Chem, Kemco form battery recycling JV. *Recycling Today* (2022).
- 4 Bittle, J. The race to close the EV battery recycling loop. *Popular Science* (2022).
- 5 Vega, S. Huayou Cobalt commits to battery recycling efforts. *The Assay* (2023).

Reviewer: 4. *The authors claimed that shipping is a minor factor for cost and CO₂, but transport of spent batteries is a different story. The authors still needs to address this point.*

Our response: In our last round of revision we have incorporated the suggestions from the reviewer (now lines 590–593 and 598–599), emphasizing the importance of the considerations concerning safety and additional costs in transporting spent batteries:

*“Business models for collection of all LIB types and sizes will likely vary from manufacturer-led to municipal or private collection programs, and can be influenced by the safety costs while collecting ~~used~~ **end-of-life** batteries as hazardous wastes.”*

*“~~Recyclable battery packing~~ **Employing reusable battery packs can save** ~~reduce refinement energy and chemical inputs~~ **used in refinement**, and **manufacturing** ~~designing~~ battery cells and*

modules favorable for extraction and integration will lower the environmental impacts of the upstream steps.”

We well understood the reviewer’s concern, but note that the current study is focused on *environmental impacts*, with emphasis on the *refinement* step. Examining cost considerations in the circular battery supply chain warrants additional investigations and is beyond the scope of the current study.

We also note that the referenced transport embedded CO₂-eq can significantly vary due to the different power sources employed, as we analyzed in **Fig. 5c** in the manuscript, which likely partially explains the different results from other ongoing studies.

Reviewer: *5. The current manuscript, after revision, is an assembly of many pieces and some of them are a little out of scope for recycling and supply chain of batteries. It gives an impression like a working report instead of a tight/coherent analysis around a central topic. Some of the less relevant context might not be necessary and can be removed to make this manuscript tight and lean.*

Our response: We thank the reviewer for suggestions on improving the readability of the manuscript, and provide reconciled responses and revisions based on the similar Question 2 by the reviewer. We politely argue that the presentations were organized around our central topic. In our study, we conducted life cycle comparison of industrial-scale lithium-ion battery recycling and mining supply chains. While the whole supply chains for battery manufacturing include refinement, extraction, and transport, we focused on the *environmental impacts* of the *refinement* step and presented our major analyses around it, investigating different important aspects of refinement. We subsequently presented studies on the complementary upstream steps (i.e., extraction and transport), and considerations on the supply chains.

To better outline our narrative, we summarized the organization of Results section in the manuscript for the reference of the reviewer:

Paragraph number	Title or main contents	Data presentation	Relevance to the main topic
1	Introductory to refinement analysis	Fig. 2	Refinement step: preparatory description of analysis pathways
2–5	“Refining lithium-ion batteries into battery-grade materials exhibited lower environmental impacts than production from mined natural materials”	Fig. 3	Refinement step: presenting overall environmental impacts
6–8	“Formats of refinement products influenced environmental impacts”	Fig. 4	Refinement step: comparing product formats, informing the fair product formats for following analyses

9–10	“Electricity consumption dominated the environmental impacts of lithium-ion battery circular refinement”	Fig. 5	Refinement step: prioritizing contributions to the refinement step
11–12	“Environmental impacts of material extraction and transport were significantly lower in circular lithium-ion battery supply chains than in conventional supply chains”	Fig. 6	Upstream steps (beyond refinement): modeling the extraction and transport steps
13	“The refinement step dominated environmental impacts of circular and conventional supply chains”	Fig. 6	Refinement + upstream steps: further analyzing the roles of refinement in the whole supply chains

To further improve the conciseness and coherence, we revised Introduction (lines 54–67, 121–122, and 132–134), Results (lines 158–159, 171–175, 301–302, 343–344, and 413–414), and Discussion (lines 501–503, and 578–584) in the new manuscript:

“Factors central to the success of recycling include the ease cost of collecting products, the cost of recycling processes, and the economic value of recovered materials. ~~The average embodied economic values of representative LIBs between 2018–2021 are shown in Fig. 1a (complete references are listed in Supplementary Information).~~ Considering in LIBs LIB prices between 2018–2021, Li, Ni, and Co comprise the highest embodied economic value (Fig. 1a, complete references are listed in Supplementary Information), and Al and Cu account for a significant weight percentage of EV battery packs (approximately 25%)⁵. While 99% of the lead-acid batteries are recycled in the U.S., LIBs exhibit ~~Despite an embodied economic value that is 2–10 times higher economic values but are compared to the lead in lead-acid batteries, only recycled 2–47% of LIBs are recycled globally⁶, compared to 99% for lead-acid batteries in the U.S. Regardless, the untapped potential of~~ The environmental benefits of circularity also strongly motivate LIB recycling constitutes a significant economic and given the vast LIB production and emission-intensive mining refinement for key constituent metals. There is a critical need to evaluate the environmental opportunity that requires evaluation across several application scales...”

“In this study we quantified the cradle-to-gate environmental impacts of battery-grade cathode material salts manufactured in conventional and circular supply chains across three major steps: material extraction, transport, and refinement (Fig. 1b), focusing on the refinement step.”

“~~Second, for the first time we~~ This use of industrial-scale recycling quantitatively identified the roles dominant role of input grid electricity in circular refinement at the balancing area level on comprehensive environmental impact metrics based on industrial-scale LIB recycling data.”

“In this study, analyses of environmental impacts were presented with a focus on the refinement step, followed by analysis of the upstream material extraction and transport steps. In LIB supply chains, the refinement step converts the collected feedstocks into battery-grade salts for further manufacturing (Fig. 2a).”

“Target products of the conventional and circular pathways were based on the GREET model and practical recycling operations, respectively. In the following sections, the overall refining environmental impacts were first analyzed, followed by influences of product formats on the refinement step and key contributors to the refining environmental impacts. Finally, upstream environmental impacts were analyzed, and compared to the refinement step. Unless specifically noted, all major analyses were based on 2021 data (data reference years were summarized in Table SI6).”

“Product format is an important factor in understanding and properly comparing LIB refinement pathways (Fig. 2). Ni and Co are key elements for battery manufacturing, and can be traded in the format of mixed metal salts or discrete salt products between battery refiners and battery manufacturers^{28,29}. ”

“To further understand the performance limiting factors in the refinement step, ~~The~~ the relative environmental impacts of input consumables (e.g., energy, water, commodity chemicals) in the gate-to-gate refinement processes were disaggregated in Fig. 5 (additional criteria air pollutants in Tables S2–S3, Figs. S4–S5).”

“Before the refinement step, LIBs undergo the ~~Upstream of gate-to-gate refinement are~~ upstream are material extraction and transport to refinement facilities (Fig. 1b).”

“Various important aspects of the environmental impacts in the refinement step were focused on and ~~Practical LIB feedstock and refinement pathways were~~ analyzed using practical data from a recycling company (~~Redwood Materials~~), and modeling was employed to examine the environmental impacts of upstream material extraction and transport steps.”

~~While the current cradle-to-gate study is focused on Li, Ni, and Co as the major output materials, the potential benefits of extracting additional LIB constitutive elements from ore (e.g., Cu and Co in Cu-Co sulfides) or from LIBs (e.g., Cu or Mn) warrants further investigation. Additionally, the same mixed-stream LIB feedstocks consumed at Redwood Materials were used to quantify NCA- and LCO-equivalent values, and results would vary for single-stream LIB feedstocks. Generally, the incremental benefits of extracting additional critical materials from concentrated sources like LIBs can offset the environmental impacts of both supply chains.”~~

■ REVIEWER 3

Reviewer: *Review Summary: This paper presents a comparative life cycle analysis for primary and secondary supply chains for lithium-ion batteries. As stated in my previous review, the paper remains well written, the topic is of interest, the core methodology is sound, and outcomes relevant for advancing the conversation around designing circular supply chains for lithium-ion batteries.*

Our response: We thank the reviewer for the favorable reception of the revised manuscript.

Reviewer: *After reviewing the author responses to my comments and updates to the manuscript, I think that:*

1) *The authors have adequately addressed the issues and questions I raised and have consequently updated the manuscript.*

2) *The updates to the manuscript have added clarity and context for the presented results, and clarified the nature and uniqueness of the main contributions from this work.*

3) *While I understand the sensitivities around industrial data, it remains a pity that this constrains transparency and limits the ability for benchmarking across different analyses, as well as drawing certain broader conclusions from reported outcomes. That said, I agree that this represents a major step in the right direction.*

4) *I believe this study makes an important contribution to foundational conversations around circular supply chains for battery. There is certainly more work required in terms of informing policy and strategy for incentivizing circular battery supply chains. It will be good to see analyses that incorporate adequate projected scale of circular supply chains, as well as economics of battery recycling, and quantitatively assess the potential impact (environmental, economic, social) of any number of policy interventions (e.g., producer responsibility, collection programs) and strategies (e.g. design for efficient disassembly, disaggregated siting of facilities for different recycling steps).*

5) *I recommend this submission for publication.*

Our response: We are grateful to the reviewer for the positive evaluations and recommendation. We also agree that additional work on policy and strategies are warranted to facilitate circular battery supply chains, and hope that our study can pave the path for more comprehensive and in-depth analyses for the rapidly growing battery recycling sector.

Reviewer: *Minor comment: Lines 524 – 527: “... optimal environmental impacts and product formats will elevate costs.”*

1) Is this demonstrably the case, or simply intuition based on likely design/operational interventions? I imagine certain adjustments to process conditions imply cost/environmental impact trade-offs, but it is not necessarily general.

Our response: We thank the reviewer for helping examine the rigorousness of the expression. While most of the existing literature on carbothermal reduction is based on lab-scale, understanding of the potential costs for optimizing process conditions is still lacking. We have accordingly revised our remarks (now lines 519–522) to:

“Our findings showed the promise of carbothermal reduction-based pyrometallurgy for pretreatment of LIB recycling; however, we note that additional work is warranted in controlling process conditions to achieve optimal environmental impacts and product formats without ~~will~~ ~~elevate~~ elevating the costs.”

REVIEWERS' COMMENTS

Reviewer #2 (Remarks to the Author):

The authors have done a decent job in addressing my concerns and I would support the manuscript to be accepted at this version.

Reviewer #3 (Remarks to the Author):

My earlier review comments had highlighted perceived limitations in the original draft which I believe were adequately addressed in the revision. I have provided a minor comment below for updating the abstract.

Minor Comments - Abstract: instead of stating in general terms what this study contributes - ".. provides important insights for advancing sustainable LIB supply chains.... informs optimization....", the Abstract should be explicit on what those key contributions are - e.g. what insights does it provide? what are the big-picture implications of the study outcomes.